# SIGMA: Refining Large Language Model Reasoning via Sibling-Guided Monte Carlo Augmentation

**Yanwei Ren**[1,2]   **Haotian Zhang**[1,2]   **Fuxiang Wu**[3]   **Jiayan Qiu**[4]
**Jiaxing Huang**[5]   **Baosheng Yu**[5]   **Liu Liu**[1,2*]

[1]School of Artificial Intelligence, Beihang University
[2]Hangzhou International Innovation Institute, Beihang University
[3]Shenzhen Institutes of Advanced Technology, Chinese Academy of Sciences
[4]University of Leicester   [5]Nanyang Technological University

## Abstract

Enhancing large language models by simply scaling up datasets has begun to yield diminishing returns, shifting the spotlight to data quality. Monte Carlo Tree Search (MCTS) has emerged as a powerful technique for generating high-quality chain-of-thought data, yet conventional approaches typically retain only the top-scoring trajectory from the search tree, discarding sibling nodes that often contain valuable partial insights, recurrent error patterns, and alternative reasoning strategies. This unconditional rejection of non-optimal reasoning branches may waste vast amounts of informative data in the whole search tree. We propose SIGMA (Sibling Guided Monte Carlo Augmentation), a novel framework that reintegrates these discarded sibling nodes to refine LLM reasoning. SIGMA forges semantic links among sibling nodes along each search path and applies a two-stage refinement: a critique model identifies overlooked strengths and weaknesses across the sibling set, and a revision model conducts text-based backpropagation to refine the top-scoring trajectory in light of this comparative feedback. By recovering and amplifying the underutilized but valuable signals from non-optimal reasoning branches, SIGMA substantially improves reasoning trajectories. On the challenging MATH benchmark, our SIGMA-tuned 7B model achieves 54.92% accuracy using only 30K samples, outperforming state-of-the-art models trained on 590K samples. This result highlights that our sibling-guided optimization not only significantly reduces data usage but also significantly boosts LLM reasoning.

## 1 Introduction

Scaling laws show that large language model (LLM) performance typically improves as model size and training data increase [23]. However, recent studies show that simply adding more common training data leads to diminishing returns, especially on complex reasoning tasks [38, 50]. This has led to a shift toward structured supervision, where detailed annotations help the model solve problems step by step [29, 12, 50, 55, 3]. Among these methods, chain-of-thought (CoT) explanations have proven especially effective [50], helping LLMs break down and solve reasoning tasks in a more organized way. Large-scale CoT datasets, such as DART-Math (∼590K samples) and MMIQC (∼2.3M samples), have demonstrated strong effectiveness in improving mathematical reasoning. Recent studies further show that even medium-sized models can achieve competitive performance when trained on such datasets [46, 31]. As illustrated in Figure 1, high-quality datasets can substantially enhance LLM reasoning capabilities while requiring far fewer training samples.

---

[*]Corresponding author: `liuliubh@buaa.edu.cn`.
Code is available at `https://github.com/frank130845/SIGMA`.

39th Conference on Neural Information Processing Systems (NeurIPS 2025).

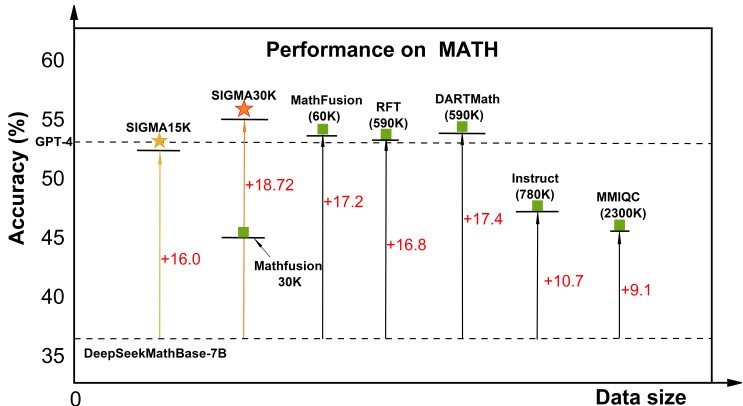

Figure 1: An illustration of the reasoning performance of fully fine-tuned DeepSeekMath-7B models on the MATH benchmark dataset. The models are trained on different datasets, with training data sizes ranging from 15K to 2300K.

However, constructing such datasets demands substantial computational resources and human effort, underscoring the need for more efficient methods to generate high-quality data. A promising direction is to automatically synthesize CoT examples using tree-based search strategies. Recently, Monte Carlo Tree Search (MCTS) has been adopted for this purpose [6], where a language model explores a branching structure by proposing and scoring candidate reasoning steps. After sufficient exploration, MCTS selects a single high-reward path as the final solution. While effective, this approach discards all sibling nodes—alternative reasoning steps that were explored but ultimately not selected. These discarded paths often contain informative signals, such as partially correct reasoning steps, recurring error patterns, and alternative reasoning strategies. Ignoring them limits the potential to extract comprehensive and structured feedback from the search process.

To address the above-mentioned issue, we introduce a new framework that reuses this lateral information to improve the selected reasoning path. We propose **SIGMA** (**SI**bling **G**uided **M**onte Carlo **A**ugmentation), which augments MCTS-based reasoning by treating sibling nodes as sources of symbolic feedback. At each step along the selected path, SIGMA compares the chosen node to its siblings to derive natural language critiques. These critiques function as approximate gradients—textual signals indicating how the reasoning step should be revised. A revision model then uses these signals to refine each step in the path. SIGMA integrates the strengths of search-based exploration and feedback-based optimization. Inspired by recent advances in symbolic supervision via LLM-generated feedback [58], it enables reasoning refinement without additional rollouts or ground-truth labels. By systematically incorporating information from all nodes produced during search, SIGMA produces higher-quality reasoning paths while maintaining computational efficiency. Experiments show that this refinement significantly improves downstream performance, offering a data-efficient alternative to large-scale dataset construction.

Our main contributions in this paper are as follows:

- We introduce **SIGMA**, a new framework that improves the selected reasoning path by incorporating sibling nodes discarded during MCTS. We aim to develop a principled data-synthesis framework that systematically exploits previously overlooked information to enhance the reasoning capabilities of larage language models.

- We leverage only the existing information from the MCTS search tree to perform symbolic optimization over reasoning paths, requiring no additional rollouts or external reward models. This design enables seamless integration into existing data generation pipelines.

- Our produced SIGMA-15K outperforms all 30K-scale baselines, and SIGMA-30K remains better or competitive with 60K-scale methods across multiple base model.Our results demonstrate the effectiveness of the proposed framework, which leverages sibling nodes to optimize reasoning paths.

## 2  Related Works

**Math Data Synthesis** Large math corpora such as WIZARDMATH and METAMATH have pushed chain-of-thought supervision into the millions, while multiple studies report that accuracy plateaus once data volume exceeds a few million examples [33, 56, 57, 5, 47]. To break this ceiling, recent work explores three complementary directions: *difficulty-aware sampling*, which allocates more generation or retention budget to problems deemed hard via pass-rate variance or confidence scores [45]; *tool-augmented generation*, which weaves external library calls into the reasoning trace so the model offloads symbolic or numeric sub-tasks [48, 16]; and *feedback-driven filtering*, which keeps only traces that satisfy verifier or preference signals, prioritising step-level quality over raw quantity [28, 30, 54, 61, 60]. Building on *difficulty-aware sampling*, **DART-Math** adapts the sampling budget on-the-fly, giving extra rejection-sampling trials to harder questions [46]; while **Math-Fusion** fuses multiple source problems into sequential, parallel, and conditional "mega-prompts" to promote relational reasoning [37]. However, these methods seldom exploit the internal structure of incorrect or discarded traces, missing opportunities to learn from intermediate reasoning failures.

**Language-Model Feedback Optimization** Recent work has begun to *close the feedback loop* by letting a model use its own outputs as training signals. The idea traces back to SELF-REFINE, where a model iteratively critiques and rewrites its answers to improve without new data [34]. DSPY generalises this to full pipelines, compiling a graph of text transformations whose prompts are automatically tuned [24]. Reflection then becomes explicit: SELF-RAG inserts special tokens so the model learns *when* to retrieve and *how* to critique its draft [2], while RL CONTEMPLATION treats the model's own evaluations as a reward, removing the need for external labels [36]. In mathematics, MATH-SHEPHERD builds a fully automatic, step-level reward pipeline that verifies and reinforces reasoning without human annotations [49]. The paradigm is unified by TEXTGRAD, which back-propagates natural-language feedback through generative pipelines ranging from code to molecular design [58]. Extensions already leverage this principle to repair code via divide-and-conquer consensus [7] and to scale automated process verifiers for math reasoning [41]. Yet most frameworks lack mechanisms to propagate fine-grained feedback across alternative reasoning paths, limiting their ability to exploit structured variation during search.

**Math Reasoning with Monte-Carlo Tree Search** As single-path CoT prompting reaches its limits, a growing body of work turns to *branching search*, letting the model explore multiple partial solutions before committing. Tree-structured exploration offers an alternative to single-chain prompting. The idea first appeared in TREE-OF-THOUGHTS, which framed reasoning as an explicit breadth- or depth-first search over "thoughts" [55]. Subsequent work replaced exhaustive search with Monte-Carlo Tree Search (MCTS): REST-MCTS* performs process-reward–guided rollouts and iteratively retrains the policy and value heads [59]; ALPHAMATH-ALMOST-ZERO adds a learned value head to steer step-level beam search without human labels [6]; MULBERRY runs *collective* MCTS across multiple models to build a 260k multimodal tree-of-reasoning corpus [53]; VERMCTS couples MCTS with a lightweight verifier to synthesise step-correct programs for proofs [4]; and preference-learning variants treat each rollout as a pairwise comparison and fine-tune via DPO [52]. While these approaches lift accuracy by sampling and selecting stronger reasoning paths, they also incur heavy inference cost and depend on auxiliary reward models or external tools, which complicates deployment. Moreover, current methods discard sibling branches after selection, underutilizing the diverse intermediate reasoning states already produced during search.

## 3  Method

In this section, we introduce the proposed SIGMA framework, a two-stage approach that integrates MCTS-based CoT data generation with sibling-level refinement to improve the quality of selected candidate reasoning paths. As shown in Figure 2, the process begins with an MCTS-based reasoning engine that explores the space of multi-step CoT paths and selects an optimal path of depth $D$ based on high-reward feedback. This selected path is then refined iteratively, one step at a time, using feedback from its sibling nodes. At each depth, we first compare the chosen node with its siblings to identify discrepancies that serve as feedback. A critique model then acts as a symbolic gradient oracle, generating directional cues in natural language rather than numerical gradients. These textual gradients are used to revise the candidate path through a textual gradient descent (TGD) process.

The following subsections detail the generation of candidate CoT paths using MCTS, the construction of sibling guidance, and the refinement of these paths based on the guidance.

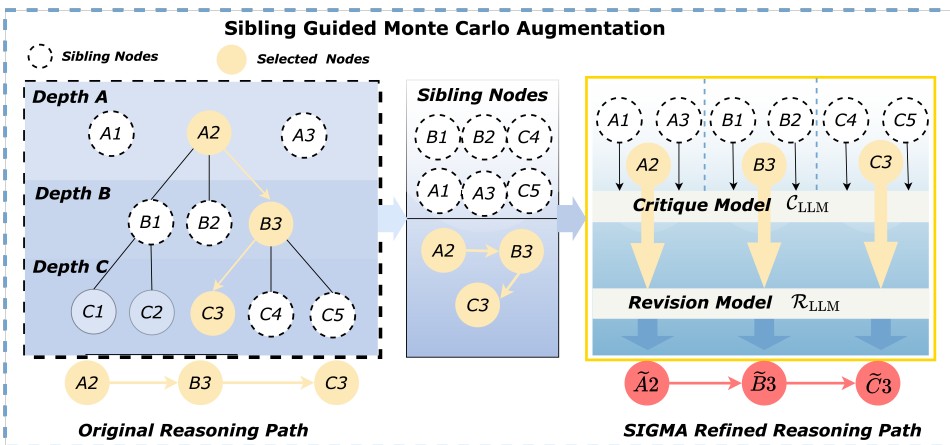

Figure 2: The main SIGMA framework for enhanced CoT data generation.

## 3.1 MCTS Reasoning Path Selection

We leverage MCTS as a tree search procedure to select a promising reasoning path. Starting from the initial state (the problem statement), MCTS iteratively builds a search tree of possible next reasoning steps (nodes), simulating outcomes and using those simulations to decide which branches to expand. We denote the finally selected path as $\mathcal{T} = \{p^{(1)}, p^{(2)}, \ldots, p^{(D)}\}$, where $p^{(d)}$ is the index of the branch chosen at depth $d \in \{1, 2, ..., D\}$ and $D$ is the largest depth. Each simulation (rollout) of MCTS reaches depth $D$, obtains a terminal reward $R$ (e.g. correctness or utility of the final answer), and then backpropagates this reward up the tree. During backpropagation, each node $n$ on the path of the simulation updates its value estimate $V_n$ to reflect the new outcome. For example, on each depth $d$, one common update is an incremental average:

$$V_n \leftarrow V_n + \frac{1}{N_n + 1}(R - V_n), \tag{1}$$

where $N_n$ is the visit count for node $n$. This recursively propagates the simulation's result to all ancestor nodes, akin to Bellman backups in dynamic programming. Additionally, MCTS uses a selection policy to balance exploration and exploitation. We adopt the Upper Confidence bound for Trees (UCT) criterion to select the child $c$ of a node $n$ by UCT:

$$c = \arg\max_{j \in \mathcal{C}(n)} \text{UCT}(n, j), \quad \text{where } \text{UCT}(n, c) = V_c + c_p \cdot \sqrt{\frac{\ln N_n}{N_c}}, \tag{2}$$

$V_c$ is the current value of the child $c$, $N_c$ denotes the counts visits of the child $c$, $\mathcal{C}(n)$ denotes the set of all child nodes of node $n$. and $c_p$ is an exploration constant. This UCT formula encourages traversal of high-value branches ($V_c$ large) while occasionally exploring less-visited siblings (via the second term). After a sufficient number of simulations, the search converges to a high-value path. Finally, MCTS selects only the top path: $\mathcal{T}^*$ (the sequence of branch choices $p^{(1:D)}$) is returned as the candidate solution CoT with highest estimated reward. We next describe how we refine this path using sibling-based feedback.

## 3.2 Sibling Guidance to Refine Reasoning Path

As illustrated in the left panel of Figure 3, To capture latent supervision from nearby alternatives, we define a symbolic loss over sibling nodes. At each depth $d \in \{1, \ldots, D\}$, we compute a discrepancy $\mathcal{L}_{\text{text}}^{(d)}$ between the selected node $p^{(d)}$ and its siblings, defined as the children of $p^{(d-1)}$ excluding $p^{(d)}$ itself. For simplicity, we omit the subscript "$(d)$". Let $T_p$ be the textual content of the selected step, and $\{T_s\}_{s \in \mathcal{S}(p)}$ the texts of its siblings, where $\mathcal{S}(p)$ denotes the set of all sibling nodes that share the same parent as the selected node $p$. This restriction to siblings under a shared parent ensures that all

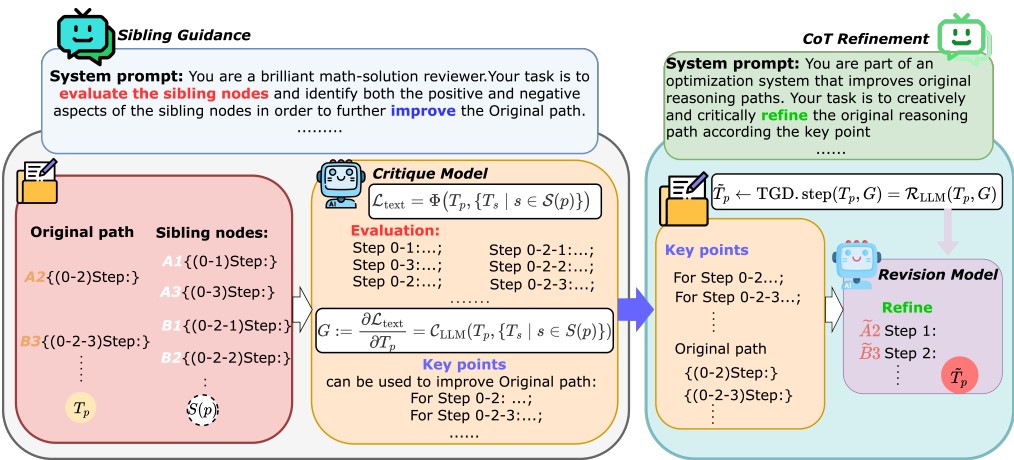

Figure 3: Visualization of the gradient computation process involving sibling nodes and the original optimal path. Node identifiers follow a hierarchical format: for instance, 0-2 denotes the second node at depth 1, and 0-2-3 denotes the third child of node 0-2 at depth 2.

candidates are conditioned on the same local context, preserving alignment in the partial reasoning path. As a result, observed differences can be reliably attributed to step-level content rather than upstream variation.

$$\mathcal{L}_{\text{text}} = \Phi\left(T_p, \{T_s \mid s \in \mathcal{S}(p)\}\right), \tag{3}$$

where $\Phi$ is a symbolic operator that compares the selected content to its siblings, which synthesizes the comparisons between the selected node $p$ and all its siblings $s \in \mathcal{S}(p)$. Unlike scalar loss values, $\mathcal{L}_{\text{text}}$ is a structured natural language output that consolidates multi-perspective judgments into a unified textual feedback. This provides a contrastive signal at each decision point, where sibling steps act as localized references that reveal potential omissions or errors.

Since $\mathcal{L}_{\text{text}}$ is non-differentiable in the conventional sense that operats over natural language rather than continuous parameters, we introduce a Critique Large Language Model $\mathcal{C}_{\text{LLM}}$ that serves as a symbolic gradient oracle. Given $(T_p, \{T_s\}_{s \in \mathcal{S}(p)})$, $\mathcal{C}_{\text{LLM}}$ produces a natural language critique that suggests how to revise $T_p$ to reduce the loss.

$$G := \frac{\partial \mathcal{L}_{\text{text}}}{\partial T_p} = \mathcal{C}_{\text{LLM}}\left(T_p, \{T_s \mid s \in \mathcal{S}(p)\}\right). \tag{4}$$

The output $G$ is not a numerical gradient, but a symbolic directional cue encoded in natural language. For instance, $\mathcal{C}_{\text{LLM}}$ may produce a message such as: "Add a justification for this computation," or "Avoid ambiguity present in sibling $s$." These critiques serve as surrogate gradients for optimizing textual reasoning through revision.

### 3.3 Revision of Original Reasoning Path

Figure 2 presents a textual example of a candidate CoT data alongside its refined version generated by the proposed SIGMA framework. Given the textual gradient $G$, we update the corresponding reasoning step by applying a single iteration of Textual Gradient Descent (TGD). In classical gradient descent, a parameter $x$ is updated as $x \leftarrow x - \eta \cdot \partial L / \partial x$, where $\eta$ is the step size. In our case, there is no explicit numerical step size or arithmetic subtraction. Instead, we employ a Revise Large Language Model $\mathcal{R}_{\text{LLM}}$ that plays the role of an optimizer. Given the current text $T_p$ and its corresponding gradient $G$ defined in Eq.(4), the revise LLM $\mathcal{R}_{\text{LLM}}$ generates an improved version $\tilde{T}_p$ intended to reduce the loss $\mathcal{L}_{\text{text}}$. This update is abstracted as $\text{TGD.step}(\cdot)$ and defined as follows:

$$\tilde{T}_p \leftarrow \text{TGD.step}\left(T_p, G\right) = \mathcal{R}_{\text{LLM}}\left(T_p, G\right). \tag{5}$$

That is, $\tilde{T}_p$ is revised based on the critique $G$. For instance, if $G$ indicates that an arithmetic step lacks justification, the revise model $\mathcal{R}_{\text{LLM}}$ will attempt to incorporate the necessary rationale, yielding a

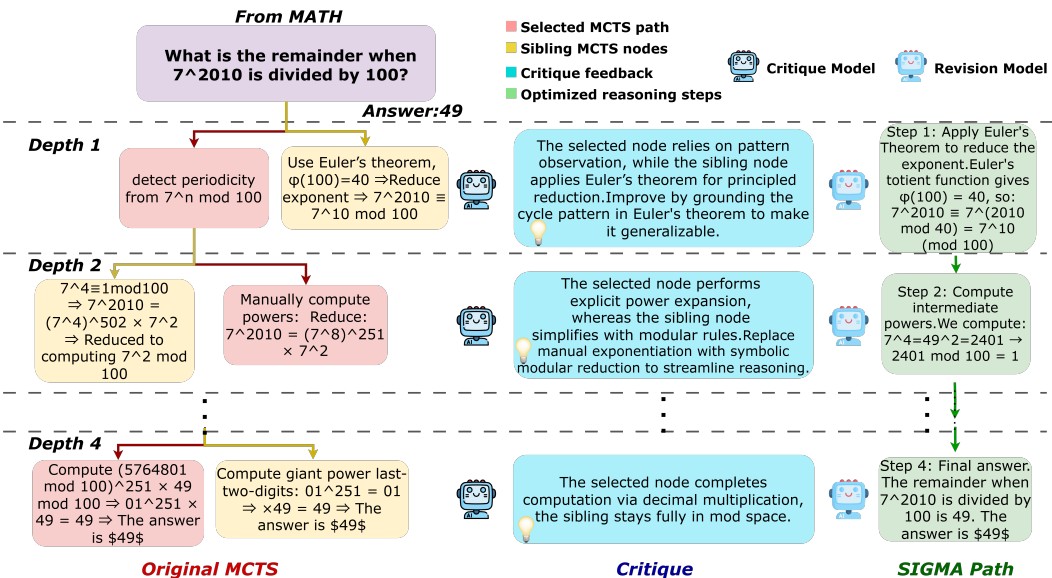

Figure 4: A textual example of the proposed SIGMA framework including three aspects: Original MCTS, critique feedback, and SIGMA path.

more complete $\tilde{T}_p$. This procedure is applied iteratively in a step-wise manner. At each depth $d$, a single textual gradient update is applied to $T_{p^{(d)}}$, while keeping all other steps fixed. Sequential updates across $d = 1$ to $D$ transform the original path $\mathcal{T}^*$ into a revised chain $\mathcal{T}^\dagger$, which exhibits greater coherence and robustness. Each update is localized and incremental, resembling a single coordinate-wise descent step in a high-dimensional optimization problem. After $D$ such updates, the procedure effectively completes one full pass of coordinate descent over the chain-of-thought.

The sibling-guided TGD process may be viewed as iterative refinement, where each reasoning step is adjusted in light of its siblings, ultimately yielding a more coherent global solution. Notably, our method operates in a model-agnostic manner and does not require access to the internal gradients or parameters of the underlying LLM. All updates are conducted in the space of natural language, mediated through $\mathcal{C}_{LLM}$ and $\mathcal{R}_{LLM}$. The final output is a refined chain-of-thought that preserves the structural strengths discovered by MCTS while improving local reasoning quality through targeted textual feedback. For instance, Figure 4 showcases the SIGMA framework's workflow using the mathematical problem: "What's the remainder when $7^{2010}$ is divided by 100". In addition, some illustrative examples are provided in Appendix F. By refining the chosen MCTS path with step-specific critique feedback, SIGMA path improved the rigor and logical consistency though shares the same answer as the original path.

## 4 Experiments

### 4.1 Experimental Setups

**Reasoning Path Selection.** We adopt the `Qwen2.5-Math-7B` as the generation model to construct search trees based on prompts from MATH [20] and GSM8K [13]. At each node, we decode $n = 3$ candidate completions and select the highest-scoring one from $k = 5$ samples, ranked by log-probability, with a maximum tree depth of 16. To enhance diversity, we generate two MCTS datasets using decoding temperatures of 0.4 and 0.7, each contributing 15K examples to form a combined 30K training set. From each search tree, we extract the path with the highest cumulative Q-value. At every depth $d$, we retain the selected node $p^{(d)}$ and up to two sibling nodes $S(p^{(d)})$ from the same parent node to construct a candidate set for step-wise refinement.

**Original Reasoning Path Refinement.** For each step, we construct a *loss&critique prompt* that includes the textual content of the selected node $T_{p^{(d)}}$ and its sibling nodes $S(p^{(d)})$. This prompt is fed into the critique model to evaluate and compare the reasoning quality among the candidates, producing a textual gradient $G^{(d)}$ that highlights potential improvements. A separate *revision prompt*

is then constructed using both $T_{p(d)}$ and $G^{(d)}$, which is passed to the revision model to generate a refined version of the reasoning step. This sibling-guided refinement is applied independently at each depth $d$, resulting in local improvements without altering the overall structure of the reasoning path. We use `GPT-4o-mini-2024-07-18` [1] as both the critique model and the revision Model.

## 4.2 Implementation Details

We fine-tune three representative base models: a math-specialized language model, DeepSeekMath-7B [42], and two general-purpose models, LLaMA3-8B [17] and Mistral-7B-v0.1 [22]. We full-finetuned and evaluated those base models on $4 \times$ H100 GPUs. To evaluate both effectiveness and generalizability, we conduct experiments using the SIGMA-refined 15K and 30K training subsets. For each model, we search for the optimal combination of learning rate and batch size, keeping all other hyperparameters fixed. We follow the official DART-Math evaluation protocol [46] and reuse their publicly released test scripts to evaluate all models under a zero-shot greedy decoding setting. For each base model, we report performance on both in-domain (ID) and out-of-domain (OOD) mathematical reasoning tasks. The ID evaluation includes GSM8K [13] and MATH [20]. The OOD evaluation include four benchmarks: CollegeMath [44], which contains 2,818 university-level problems spanning seven mathematical domains; DeepMind Mathematics [40], a curriculum-aligned suite of 1,000 problems designed for students up to age 16; OlympiadBench-Math [19], which consists of 675 Olympiad-level problems from international contests; and TheoremQA [9], which evaluates symbolic reasoning using theorem statements from various STEM disciplines.

We compare SIGMA-tuned models against a wide range of competitive baselines, including both instruction-tuned and reinforcement-optimized approaches. These include MetaMath [56], Wizard-Math [33], MMIQC [31], MathScale [44], RefAug [60], DART-Math [46] and MathFusion [37]. For DeepSeekMath-7B, we additionally include its official supervised variant [42]. Base models that are fine-tuned directly on MATH and GSM8K without augmentation are reported under the Standard setting. Most baseline results are taken from MathFusion [37] or DART-Math [46], where applicable.

## 4.3 Main Results

Table 1 summarizes the performance of base models fine-tuned on our SIGMA datasets (SIGMA-15K and SIGMA-30K), alongside models fine-tuned on large-scale datasets generated by various baseline methods.Furthermore, we present an extended analysis of performance across six different benchmarks in Appendix B, offering a complete view of the evaluation results.

**SIGMA-15K Beats All 30K Models.** With only 15K training examples, SIGMA-15K outperforms all prior methods trained on 30K samples across all three model backbones. For example, on DeepSeekMath-7B, SIGMA-15K achieves 47.0 average accuracy, exceeding the best 30K baselines such as MathFusion-DSMath-7B (45.7) and MathFusion-DSMath-7B (Sequential) (45.7). Similar trends are observed on Mistral-7B (31.6 vs. 29.9 from MathFusion) and LLaMA3-8B (36.1 vs. 35.6 from MathFusion), demonstrating SIGMA's superior data efficiency.

**LLaMA3-8B Series.** The upper part of the table presents the performance of the LLaMA3-8B model series. Controlling for identical data budgets (30K samples), our SIGMA-8B-30K model achieves an overall average of 37.7, a 2.1-point absolute gain over the MathFusion-30K baseline. Notably, SIGMA-30K outperforms MathFusion-30K on every benchmark, with the largest improvement on the DeepMind set (+5.5). Even when using only 15K samples, our SIGMA-15K model still achieves a 36.0 average, 0.4 points higher than MathFusion-30K trained with double the data. Moreover, scaling to 60K samples yields a significant boost to 40.2 average accuracy, surpassing DART-Math-60K (37.6) and establishing a new best performance within this family. These results highlight SIGMA's strong data efficiency and scalability.

**Mistral-7B Series.** The second part of Table 1 reports fine-tuning results on the Mistral-7B-v0.1 backbone. With a 30K sample budget, SIGMA-7B-30K achieves 34.9, a 5.0-point gain over MathFusion-30K (29.9), and outperforms it on every task, most notably +14.5 on DeepMind. Even using just 15K samples, SIGMA-7B-15K scores 31.6, still 1.7 points above MathFusion-30K. When scaled to 60K samples, SIGMA-7B-60K reaches 36.9, outperforming the DART-Math-60K model (32.9) and demonstrating consistent improvement with data scaling.

Table 1: Performance comparison across base models and training strategies. Results are reported as exact-match accuracy under 0-shot greedy decoding(temperature = 0). All scores were obtained from the first attempt. Arrows indicate accuracy changes relative to the baseline in blue background. Some results are quoted from MathFusion [37] and some are quoted from DART-Math [46].

| Model | # Samples | In-Domain | | Out-of-Domain | | | | AVG |
|---|---|---|---|---|---|---|---|---|
| | | MATH | GSM8K | College | DM | Olympiad | Theorem | |
| **LLaMA3-8B (General Base Model)** | | | | | | | | |
| Llama3-8B-MetaMath | 400K | 32.5 | 77.3 | 20.6 | 35.0 | 5.5 | 13.8 | 30.8 |
| Llama3-8B-RFT | 590K | 39.7 | 81.7 | 23.9 | 41.7 | 9.3 | 14.9 | 35.2 |
| Llama3-8B-DART-Math | 590K | 46.6 | 81.1 | 28.8 | 48.0 | 14.5 | 19.4 | 39.7 |
| Llama3-8B-MMIQC | 2.3M | 39.5 | 77.6 | 29.5 | 41.0 | 9.6 | 16.2 | 35.6 |
| **Llama3-SIGMA-8B-15K** | **15K** | 36.0 | **82.0**↑4.1 | 24.2 | **42.0** | 10.5 | **22.0**↑5.0 | **36.1**↑0.5 |
| **Llama3-SIGMA-8B-30K** | **30K** | 40.8↑2.0 | 79.5↑1.6 | 26.3↑0.8 | 47.5↑5.5 | 12.7↑0.1 | 19.1↑2.1 | 37.7↑2.1 |
| MathFusion (Sequential) | 30K | 38.8 | 77.9 | 25.1 | 42.0 | 12.6 | 17.0 | 35.6 |
| MathFusion (Conditional) | 30K | 34.7 | 76.9 | 21.2 | 27.4 | 11.9 | 15.5 | 31.3 |
| MathFusion (Parallel) | 30K | 38.1 | 75.4 | 25.5 | 41.9 | 11.9 | 18.9 | 35.3 |
| Llama3-8B-MetaMat | 60K | 28.7 | 78.5 | 19.7 | 31.3 | 5.3 | 16.1 | 29.9 |
| Llama3-8B-MMIQC | 60K | 24.4 | 69.7 | 13.4 | 30.9 | 5.2 | 10.6 | 25.7 |
| Llama3-8B-RefAug | 60K | 20.3 | 68.6 | 15.5 | 29.1 | 5.5 | 13.0 | 25.3 |
| Llama3-8B-DART-Math | 60K | 39.6 | 82.2 | 27.9 | 39.9 | 12.9 | 22.9 | 37.6 |
| MathFusion-Llama3-8B | 60K | 46.5 | 79.2 | 27.9 | 43.4 | 17.2 | 20.0 | 39.0 |
| **Llama3-SIGMA-8B-60K** | 60K | 44.9 | **82.4**↑3.2 | 28.1 | **49.2**↑5.8 | 15.3 | 21.3↑1.3 | **40.2**↑1.2 |
| **Mistral-7B-v0.1 (General Base Model)** | | | | | | | | |
| Mistral-7B-MetaMath | 400K | 29.8 | 76.5 | 19.3 | 28.0 | 5.9 | 14.0 | 28.9 |
| Mistral-7B-WizardMath-V1.1 | 418K | 32.3 | 80.4 | 23.1 | 38.4 | 7.7 | 16.6 | 33.1 |
| Mistral-7B-RFT | 590K | 38.7 | 82.3 | 24.2 | 35.6 | 8.7 | 16.2 | 34.3 |
| Mistral-7B-DART-Math | 590K | 45.5 | 81.1 | 29.4 | 45.1 | 14.7 | 17.0 | 38.8 |
| **Mistral-SIGMA-7B-15K** | **15K** | 30.0 | 75.3↑1.4 | 20.8↑1.9 | 39.5↑10.2 | 7.7 | **16.3**↑0.8 | **31.6**↑1.7 |
| **Mistral-SIGMA-7B-30K** | **30K** | 35.5↑2.8 | 78.6↑4.7 | 22.1↑3.2 | **43.8**↑14.5 | 11.1↑1.8 | 18.0↑2.5 | 34.9↑5.0 |
| MathFusion (Sequential) | 30K | 32.7 | 73.9 | 18.9 | 29.3 | 9.3 | 15.5 | 29.9 |
| MathFusion (Conditional) | 30K | 26.3 | 73.0 | 15.6 | 21.4 | 7.3 | 12.8 | 26.1 |
| MathFusion (Parallel) | 30K | 30.9 | 75.1 | 20.9 | 26.5 | 11.0 | 15.2 | 29.9 |
| Mistral-7B-MMIQC | 60K | 17.3 | 61.4 | 11.1 | 13.5 | 5.0 | 5.9 | 19.0 |
| Mistral-7B-RefAug | 60K | 17.4 | 63.1 | 12.5 | 18.1 | 3.9 | 11.1 | 21.0 |
| Mistral-7B-MetaMath | 60K | 22.7 | 70.8 | 14.1 | 27.2 | 5.0 | 12.2 | 25.3 |
| Mistral-7B-DART-Math | 60K | 34.1 | 77.2 | 23.4 | 36.0 | 8.7 | 18.2 | 32.9 |
| MathFusion-Mistral-7B | 60K | 41.6 | 79.8 | 24.3 | 39.2 | 13.6 | 18.1 | 36.1 |
| **Mistral-SIGMA-7B-60K** | 60K | 40.3 | 79.2 | 24.1 | 46.1↑6.9 | 12.3 | 19.2↑1.1 | 36.9↑0.8 |
| **DeepSeekMath-7B (Math-Specialized Base Model)** | | | | | | | | |
| DeepSeekMath-7B-RFT | 590K | 53.0 | 88.2 | 41.9 | 60.2 | 19.1 | 27.2 | 48.3 |
| DeepSeekMath-7B-DART-Math | 590K | 53.6 | 86.8 | 40.7 | 61.6 | 21.7 | 32.2 | 49.4 |
| DeepSeekMath-7B-Instruct | 780K | 46.9 | 82.7 | 37.1 | 52.2 | 14.2 | 28.1 | 43.5 |
| DeepSeekMath-7B-MMIQC | 2.3M | 45.3 | 79.0 | 35.3 | 52.9 | 13.0 | 23.4 | 41.5 |
| **DeepSeekMath-SIGMA-7B-15K** | **15K** | 52.2↑2.3 | 81.1↑4.5 | 37.4 | 64.5 | 20.3 | **26.1**↑3.3 | **47.0**↑1.3 |
| **DeepSeekMath-SIGMA-7B-30K** | **30K** | 54.9↑5.0 | 82.2↑5.6 | 36.7 | 67.2↑2.6 | 21.6 | 26.6↑3.8 | 48.2↑2.5 |
| MathFusion (Sequential) | 30K | 49.9 | 76.6 | 38.8 | 64.6 | 21.6 | 22.8 | 45.7 |
| MathFusion (Conditional) | 30K | 48.5 | 74.6 | 37.0 | 55.2 | 19.3 | 19.0 | 42.3 |
| MathFusion (Parallel) | 30K | 50.9 | 76.7 | 38.9 | 62.2 | 19.0 | 23.8 | 45.3 |
| DeepSeekMath-7B-MMIQC | 60K | 26.3 | 60.6 | 19.2 | 41.5 | 10.4 | 6.8 | 27.5 |
| DeepSeekMath-7B-RefAug | 60K | 33.1 | 71.6 | 26.2 | 35.4 | 10.5 | 14.0 | 31.8 |
| DeepSeekMath-7B-MetaMath | 60K | 40.0 | 79.0 | 33.2 | 45.9 | 9.5 | 18.9 | 37.8 |
| DeepSeekMath-7B-DART-Math | 60K | 51.4 | 82.9 | 39.1 | 62.8 | 21.0 | 27.4 | 47.4 |
| MathFusion-DeepSeekMath-7B | 60K | 53.4 | 77.9 | 39.8 | 65.8 | 23.3 | 24.6 | 47.5 |
| **DeepSeekMath-SIGMA-7B-60K** | 60K | **56.5**↑3.1 | **81.7**↑3.8 | 37.2 | **68.4**↑2.6 | 22.5 | **29.3**↑4.7 | **49.3**↑1.8 |

Table 2: Ablation study on 15K datasets. Results are reported as exact-match accuracy under 0-shot greedy decoding(temperature = 0). SIGMA outperforms both the unrefined MCTS best-path and GPT-4o-mini CoT (Blackbox) baselines under identical training settings.

| Model | # Samples | MATH | GSM8K | College | DM | Olympiad | Theorem | AVG |
|---|---|---|---|---|---|---|---|---|
| DeepSeekMath-7B-Blackbox | 15K | 50.5 | 77.3 | **38.6** | 62.0 | 19.1 | 25.8 | 45.5 |
| DeepSeekMath-7B-MCTS | 15K | 40.7 | 58.3 | 30.2 | 43.7 | 18.1 | 10.7 | 33.6 |
| **DeepSeekMath-SIGMA-7B-15K** | 15K | **52.2**↑1.7 | **81.1**↑3.8 | 37.4↓1.2 | **64.5**↑2.5 | **20.3**↑1.2 | **26.1**↑0.3 | **46.9**↑1.4 |
| Llama3-8B-Blackbox | 15K | 34.5 | 77.9 | 21.1 | 39.9 | 11.1 | 16.6 | 33.5 |
| Llama3-8B-MCTS | 15K | 27.8 | 40.2 | 18.3 | 34.3 | **12.6** | 8.5 | 23.6 |
| **Llama3-SIGMA-8B-15K** | 15K | **36.0**↑1.5 | **82.0**↑4.1 | **24.2**↑3.1 | **42.0**↑2.1 | 10.5↓2.1 | **22.0**↑5.4 | **36.1**↑2.6 |
| Mistral-7B-Blackbox | 15K | **31.2** | 68.1 | 20.6 | 37.7 | 10.5 | 6.1 | 29.0 |
| Mistral-7B-MCTS | 15K | 21.7 | 45.6 | 20.1 | 25.4 | **11.1** | 8.3 | 22.0 |
| **Mistral-SIGMA-7B-15K** | 15K | 30.0↓1.2 | **75.3**↑7.2 | **20.8**↑0.2 | **39.5**↑1.8 | 7.7↓3.4 | **16.3**↑8.0 | **31.6**↑2.6 |

Table 3: Comparison of different Critique-and-Revision (C&R) configurations under 15K training samples. Results are reported as pass@1 exact-match accuracy under 0-shot greedy decoding (temperature = 0). (C&R) denotes using the model as a Critique and Revision model.

| C&R Model / Method | MATH | GSM8K | College | DM | Olympiad | Theorem | AVG |
|---|---|---|---|---|---|---|---|
| **Llama3-8B Series** | | | | | | | |
| Qwen2.5-7B-Instruct (C&R) | 35.3 | 77.1 | 21.7 | 39.5 | 9.6 | 16.9 | 33.3 |
| Qwen2.5-72B-Instruct (C&R) | 36.3 | 78.9 | 23.2 | 39.4 | 9.8 | 18.0 | 34.3 |
| GPT4o-mini (C&R) | 36.0 | 82.0 | 24.2 | 42.0 | 10.5 | 22.0 | 36.1 |
| MathFusion (30K Avg.) | 37.2 | 76.7 | 23.9 | 37.1 | 12.1 | 17.1 | 34.1 |
| **Mistral-7B-v0.1 Series** | | | | | | | |
| Qwen2.5-7B-Instruct (C&R) | 28.8 | 79.3 | 18.4 | 33.3 | 6.7 | 13.5 | 30.0 |
| Qwen2.5-72B-Instruct (C&R) | 30.6 | 76.8 | 21.1 | 35.3 | 6.8 | 15.6 | 31.0 |
| GPT4o-mini (C&R) | 30.0 | 75.3 | 20.8 | 39.5 | 7.7 | 16.3 | 31.6 |
| MathFusion (30K Avg.) | 30.0 | 74.0 | 18.5 | 25.7 | 9.2 | 14.5 | 28.6 |
| **DeepSeekMath-7B-Base Series** | | | | | | | |
| Qwen2.5-7B-Instruct (C&R) | 49.8 | 80.6 | 38.3 | 60.6 | 18.4 | 24.1 | 45.3 |
| Qwen2.5-72B-Instruct (C&R) | 51.3 | 81.9 | 39.9 | 64.8 | 17.5 | 23.8 | 46.5 |
| GPT4o-mini (C&R) | 52.2 | 81.1 | 37.4 | 64.5 | 20.3 | 26.1 | 47.0 |
| MathFusion (30K Avg.) | 49.8 | 76.0 | 38.2 | 60.7 | 20.0 | 21.9 | 44.4 |

**DeepSeekMath-7B Series.** Our DeepSeekMath-SIGMA-7B-30K model achieves a 48.2 average across six benchmarks, 2.5 points higher than MathFusion-30K (45.7) and even 0.7 points above MathFusion-DSMATH-7B trained on 60K samples (47.5). SIGMA-7B-15K still attains 47.0, 1.3 points above MathFusion-30K. Further scaling to 60K examples yields 49.7 average accuracy, setting a new state-of-the-art among all tested models. These results show that with just 30K–60K synthetic examples, our SIGMA pipeline not only outperforms all MathFusion variants regardless of data budget, but also scales effectively with larger synthetic datasets.

Overall, with a comparable or smaller number of tokens(Detailed token-level statistics are provided in Appendix D), SIGMA datasets yield consistent and significant improvements across all model backbones and data budgets. Even small-scale SIGMA sets (15K) outperform large-scale baselines (30K–60K), while scaling to 60K examples further extends the advantage, highlighting the robustness and scalability of our synthetic data generation strategy. Beyond mathematical reasoning tasks, we further evaluate the applicability of the SIGMA framework on non-mathematical domains. As reported in Appendix C, the results demonstrate that our approach exhibits strong generality across diverse tasks.

## 4.4 Ablation Studies

To better assess the impact of SIGMA's sibling-guided refinement, we conduct two controlled ablation experiments across all three base models: DeepSeekMath-7B, LLaMA3-8B, and Mistral-7B-

v0.1. In all settings, the number of training samples is fixed at 15K, and training configurations remain identical to SIGMA-15K.

**Comparison to vanilla MCTS paths.** We construct a baseline by extracting the selected best path from each MCTS tree used in SIGMA-15K, without applying any refinement. These paths are directly used as training targets, forming the MCTS-15K dataset. As shown in Table 2, SIGMA significantly outperforms vanilla MCTS across all models: +13.3 on DeepSeekMath-7B (46.9 vs. 33.6), +12.5 on LLaMA3-8B (36.1 v.s 23.6), and +9.6 on Mistral-7B (31.6 v.s 22.0). These improvements highlight SIGMA's ability to densify the learning signal from existing MCTS trees without additional generation.

**Comparison to black-box CoT generation.** To further isolate the effect of sibling-based refinement, we compare our SIGMA-15K against BLACKBOX-15K, a dataset generated by prompting `GPT-4o-mini-2024-07-18` with a standard prompt to obtain Chain-of-Thought (CoT) outputs for each problem in the SIGMA-15K query set. Despite sharing the same generator model, SIGMA consistently outperforms black-box generation on DeepSeekMath (+1.4), LLaMA3 (+2.6), and Mistral (+1.6) in average accuracy. Notably, SIGMA achieves this without relying on full-solution sampling, instead leveraging internal structural comparisons among sibling nodes to guide refinement.

**Replacing the Teacher Model.** To examine the generality of SIGMA's Critique-and-Revision (C&R) pipeline, we replace the teacher model used in refinement with several alternatives, including `Qwen2.5-7B-Instruct`, `Qwen2.5-72B-Instruct`, and `GPT-4o-mini`. As reported in Table 3, all teacher models substantially enhance student performance compared to the MathFusion (30K Avg.) baseline, even under the same 15K data budget. Notably, `GPT-4o-mini (C&R)` achieves the strongest results across all three backbones, reaching 47.0 average accuracy on DeepSeekMath-7B, 36.1 on LLaMA3-8B, and 31.6 on Mistral-7B. These findings indicate that SIGMA's refinement process is highly transferable: larger or more capable teachers yield proportionally better student performance, yet even smaller open-weight models (e.g., Qwen2.5-7B) maintain strong gains. This demonstrates that SIGMA's C&R mechanism is not tied to a specific teacher model and can generalize across instruction-tuned LLMs with varying scales.

These results confirm that SIGMA enhances the supervision quality of MCTS data by extracting and reusing informative contrastive signals from partial rollouts. Even under identical data budgets and generator capabilities, SIGMA yields superior downstream performance.

# 5 Conclusion

In this work, we focus on enhancing the utility of MCTS-generated reasoning traces by revisiting the discarded sibling nodes. We propose **SIGMA**, a refinement framework that leverages sibling comparisons and gradient-like language model feedback to improve the quality of step-by-step reasoning data. By applying SIGMA to MCTS paths, we construct compact yet high-quality datasets (SIGMA-15K,SIGMA-30K and SIGMA-60K), which consistently outperform much larger baselines across multiple base models. Despite using only one-fourth to one-half the training volume of typical instruction-tuning datasets, SIGMA consistently improves mathematical reasoning performance across model families and task domains.

**Limitations**. While our framework demonstrates promising results, several limitations remain. First, the use of GPT-4o-mini as the backbone model may restrict overall performance due to its relatively limited capabilities compared to larger models. Second, we focus on full fine-tuning of 7B-scale models and do not explore larger models, which could potentially yield stronger results.

# Acknowledgments

This work was supported by the National Natural Science Foundation of China (62441617, KZ37132301, KZ37145801), and the Research Funding of Hangzhou International Innovation Institute of Beihang University (Grant No.2024KQ095).

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

# A  Evaluation Result on Qwen2.5-Math-7B

Table 4: Comparison of SIGMA and other math reasoning models based on Qwen2.5Math-7B. Results are reported aspass@1 exact-match accuracy under 0-shot greedy decoding (temperature = 0) across six mathematical reasoning benchmarks.

| Model | # Samples | MATH | GSM8K | College | DM | Olympiad | Theorem | **AVG** |
|---|---|---|---|---|---|---|---|---|
| Rstar@Greedy | 747K | 78.40 | 89.70 | 52.50 | – | 47.10 | – | – |
| Eurus-2-7B-PRIME | 230K | 53.88 | 42.99 | 20.97 | 47.30 | 22.22 | 18.62 | 34.33 |
| OpenMath-Nemotron-7B | 3.2M | 40.52 | 73.39 | 32.93 | 54.70 | 11.41 | 13.63 | 37.76 |
| qwen2.5math-instruct | 2.5M | 82.16 | 95.15 | 46.80 | 82.00 | 41.60 | 40.75 | 64.74 |
| AceMath-7B-Instruct | 2.3M | 75.40 | 58.38 | 45.82 | 59.70 | 39.85 | 30.75 | 51.65 |
| DART-Math-60K | 60K | 63.14 | 91.28 | 47.23 | 74.90 | 26.81 | 35.38 | 56.46 |
| MathFusion-60K | 60K | 56.14 | 59.59 | 37.08 | 67.70 | 41.19 | 27.87 | 48.26 |
| **SIGMA-60K (ours)** | 60K | **79.92** | **89.23** | **45.17** | **88.80** | **43.41** | **47.00** | **65.59** |

To further verify the broad applicability of the SIGMA framework beyond DeepSeek-Math-7B, we conduct additional experiments on the Qwen2.5-Math-7B model. As shown in Table 4, SIGMA-60K achieves an average accuracy of 65.6, outperforming all baseline models by clear margins across six benchmarks.

Compared with Rstar[18], SIGMA-60K achieves similar results on MATH and GSM8K while showing stronger robustness on more complex tasks such as DeepMind and Theorem. Relative to Eurus-2-7B-PRIME[14] and OpenMath-Nemotron-7B[10], SIGMA surpasses them by +31.3 and +27.8 points in average accuracy respectively, despite using far fewer training samples. When compared to the Qwen2.5math-instruct base model, SIGMA-60K still gains +0.9 overall, with particularly large improvements on DeepMind (+6.8) and Theorem (+6.2). Over AceMath-7B-Instruct[32], the average gain reaches +13.9, showing consistent advantage across all tasks. Furthermore, SIGMA-60K outperforms both DART-Math-60K and MathFusion-60K by +9.1 and +17.3 in average accuracy, respectively, achieving the best results on every individual benchmark.

Overall, these results confirm that SIGMA's improvements are not limited to a specific base model: even when applied to Qwen2.5Math-7B, the framework maintains strong generalization and delivers substantial gains across diverse mathematical reasoning tasks.

# B  Analysis of Performance Across Six Different Benchmarks

In Appendix B, We provide a benchmark level breakdown across six diverse mathematical reasoning tasks (GSM8K[13], MATH[20], College[44], DeepMind[40], Olympiad[19], and Theorem[9]), highlighting SIGMA's robustness and its ability to generalize across varying levels of problem difficulty.

To systematically quantify how each data construction technique impacts performance across our six benchmarks:GSM8K, MATH, College, DeepMind, Olympiad, and Theorem.We present different benchmarks' analysis comparing SIGMA with MetaMath [56], MMIQC [31], RefAug [60], DART-Math, and MathFusion (Figure 5). We highlight both absolute and relative gains, as well as consistency across the three backbone models, to demonstrate SIGMA's robustness and general applicability.

**60K-Sampled Datasets Comparsion Across Six Benchmarks.** *GSM8K:* SIGMA consistently outperforms MathFusion across all base architectures. Furthermore, it matches the performance of DART-Math, and remarkably, all three models converge to nearly identical accuracies when fine-tuned with our method. *MATH:* On the more demanding MATH benchmark, SIGMA delivers substantial gains over DART-Math for every backbone. In particular, it also exceeds MathFusion's results on both LLaMA and DeepSeekMath, underscoring its robustness on complex arithmetic reasoning. *College:* In the College evaluation, SIGMA yields pronounced improvements for Mistral and DeepSeekMath. This demonstrates the strength of our data generation in enhancing performance on intermediate-difficulty problems. *DeepMind:* SIGMA outstrips all competing approaches—including DART-Math and MathFusion—by a wide margin on DeepMind. The consistent uplift across every model highlights its effectiveness on advanced reasoning tasks. *Olympiad:*

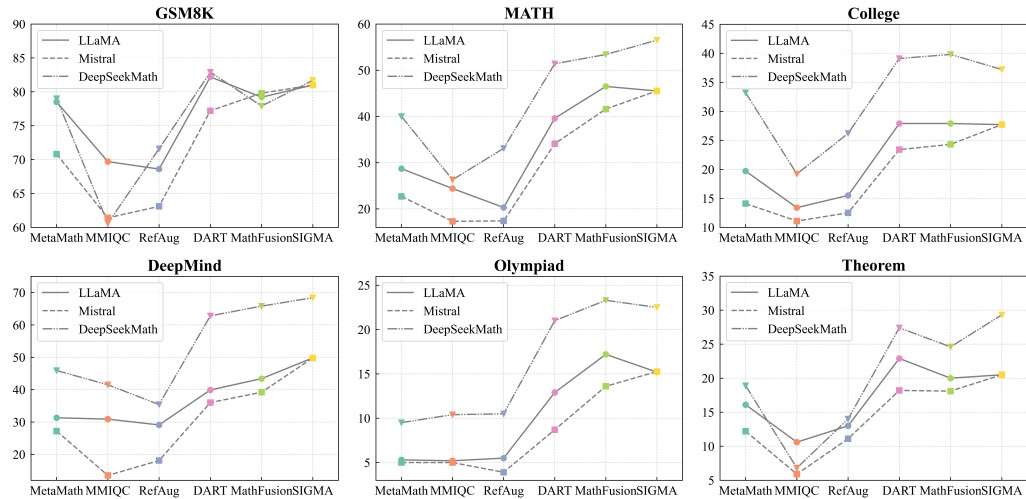

Figure 5: Performance comparison of models fine-tuned on **60k**-sample datasets generated by different methods, evaluated across six benchmark tasks. Different colored dots represent different methods.

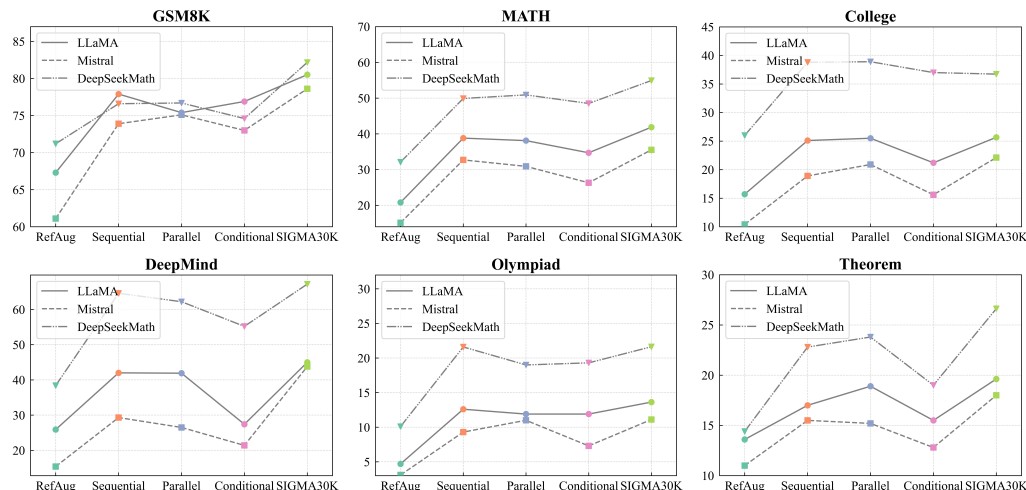

Figure 6: Performance comparison of models fine-tuned on **30k**-sample datasets generated by different methods, evaluated across six benchmark tasks. Different colored dots represent different methods.

For the Olympiad benchmark, SIGMA falls slightly short of MathFusion on LLaMA and Mistral but outperforms it on DeepSeekMath. Crucially, it still achieves clear, uniform improvements over DART-Math across all three architectures. *Theorem:* On Theorem, SIGMA secures a clear advantage over MathFusion, confirming its ability to generate data that strengthens formal and symbolic reasoning.

**30K-Sampled Datasets Comparsion Across Six Benchmarks.** As shown in Figure 6, *GSM8K:* SIGMA surpasses all baselines on GSM8K when fine-tuned with 30 K samples, demonstrating consistent superiority across multiple base models. *MATH:* On MATH benchmark, SIGMA delivers marked gains over every competing method, showing stable uplift across different base models. *College:* In College evaluations, SIGMA yields notable improvements for each architecture, demonstrating that our data construction effectively enhances general-purpose models' ability for reasoning challenging mathematical problems. *DeepMind:* SIGMA delivers substantial gains over MathFusion on DeepMind tasks, underscoring its strong adaptability to DeepMind dataset. *Olympiad:* For Olympiad problems, SIGMA maintains consistent gains over DART-Math across all backbones and

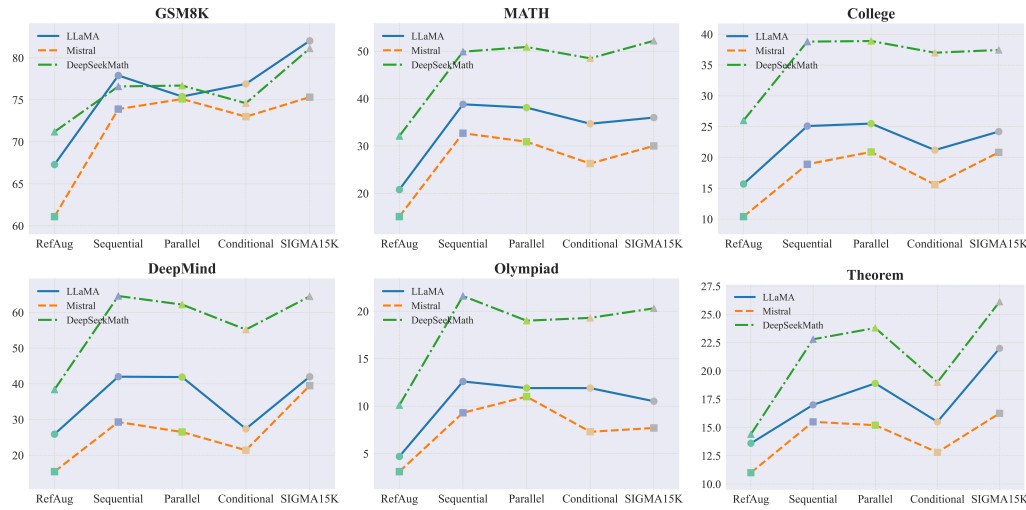

Figure 7: Performance comparison of models fine-tuned on **15k**-sample datasets generated by our SIGMA15k and **30k**-sample datasets generated by other methods, evaluated across six benchmark tasks. Different colored dots represent different methods.

narrows the gap with MathFusion, confirming robust performance on competition-level questions. *Theorem:* On the Theorem dataset, SIGMA achieves uniform improvements over every baseline, illustrating that its advantages in theorem reasoning persist regardless of dataset size.

**SIGMA-15K vs. 30K-Scale Methods: Performance Across Six Benchmarks.** As shown in Figure 7, *GSM8K:* With 15K training samples, SIGMA still leads all augmentation strategies on GSM8K, delivering uniformly higher accuracies across LLaMA, Mistral and DeepSeekMath and narrowing performance disparities among them. *MATH:* Even at reduced scale, SIGMA achieves clear improvements on MATH, outperforming each baseline and preserving its edge on all three backbones. *College:* In the College evaluation, SIGMA demonstrates a pronounced advantage, boosting each architecture's capability to tackle intermediate-difficulty tasks despite the limited training data. *DeepMind:* SIGMA adapts effectively to the DeepMind dataset, surpassing Math-Fusion and other approaches across all backbones, which underscores its resilience on complex reasoning challenges. *Olympiad:* On Olympiad questions, SIGMA outperforms DART-Math for every base model and approaches MathFusion's standard, confirming its steady competitiveness on high-difficulty problems even at reduced scale. *Theorem:* For the Theorem benchmark SIGMA sustains its lead over all competing strategies, demonstrating that its strengths in reasoning persist when trained on only fifteen thousand instances.

## C Evaluation on Commensense Reasoning Tasks

To further evaluate the generalization capability of the SIGMA method on text-based reasoning tasks, we compared the performance of SIGMA framework against several baselines (Zero-shot, SKD[51, 27], Distill Step-by-Step[21], Rephrase Question[56], Question Aug[26], Answer Aug[56], and Revthink[8]) on three commonsense reasoning datasets: CommonsenseQA (CSQA)[43], StrategyQA (SQA)[15], and ARC-Challenge (ARC)[11]. Following the experimental setup of RevThink, we fine-tune Mistral-7B-Instruct-v0.3 and Gemma-7B-Instruct using the LoRA with data generated by SIGMA. All results are reported as pass@1 exact-match accuracy under 0-shot greedy decoding (temperature = 0). Detailed results are presented in Table 5.

As clearly shown in Table 5, the SIGMA method significantly outperforms all baseline methods across all three datasets (SQA, CSQA, ARC) and also achieves the highest average accuracy on both Mistral-7B-Instruct-v0.3 and Gemma-7B-Instruct models. Specifically, on Mistral-7B, SIGMA achieves an average accuracy of 75.74%, showing a noticeable improvement over the second-best method, Revthink (+0.66). The advantage of SIGMA is even more pronounced on Gemma-7B, where it reaches an average accuracy of 73.63%, considerably surpassing Revthink(+2.36).

Table 5: Performance comparison of different reasoning methods on Mistral-7B-Instruct-v0.3 and Gemma-7B-Instruct. Results are reported as pass@1 exact-match accuracy under 0-shot greedy decoding (temperature = 0).

| Methods | Mistral-7B-Instruct-v0.3 | | | | Gemma-7B-Instruct | | | |
| --- | --- | --- | --- | --- | --- | --- | --- | --- |
| | SQA | CSQA | ARC | AVG | SQA | CSQA | ARC | AVG |
| Zero-shot | 53.89 | 62.57 | 73.68 | 63.38 | 56.33 | 66.26 | 68.34 | 63.64 |
| SKD | 63.76 | 71.86 | 74.66 | 70.09 | 56.77 | 72.48 | 73.29 | 67.51 |
| Distill Step-by-Step | 64.19 | 71.92 | 75.32 | 70.48 | 56.77 | 73.01 | 72.92 | 67.57 |
| Rephrase Question | 65.07 | 70.19 | 74.51 | 69.92 | 54.15 | 70.22 | 72.37 | 65.58 |
| Question Aug | 65.07 | 72.23 | 73.32 | 70.21 | 57.21 | 68.11 | 72.74 | 66.02 |
| Answer Aug | 66.38 | 69.12 | 76.77 | 70.76 | 57.21 | 73.01 | 73.92 | 68.05 |
| Revthink | 70.97 | 75.76 | 78.50 | 75.08 | 64.19 | 74.53 | 75.09 | 71.27 |
| **SIGMA (Ours)** | **71.62** | **76.30** | **79.30** | **75.74** | **67.40** | **75.40** | **78.10** | **73.63** |

These results strongly demonstrate the effectiveness of the SIGMA method. Fine-tuning with our proposed approach significantly enhances the models' performance on purely text-based commonsense reasoning tasks, validating the method's strong generalization ability across different models and tasks.

## D Computation and Data Efficiency Analysis

Table 6: Comparison of synthesis costs for different methods. GPU cost, API cost, and total cost are measured in USD.

| Method | GPU Type | GPU Hours | Price | API Cost | Total Cost |
| --- | --- | --- | --- | --- | --- |
| DART-Math | A100 | 3840 | 0.9 | – | 3456 |
| SIGMA (GPT-4o-mini as C&R) | RTX 4090 | 168 | 0.35 | 47.6 | 106.4 |
| SIGMA (Qwen2.5-7B-Instruct as C&R) | RTX 4090 | 200 | 0.35 | – | 70.0 |

We further analyze the synthesis cost and data efficiency of the SIGMA framework. As a data-efficient approach, maintaining low synthesis and computational cost is central to SIGMA's practicality and scalability. Unlike previous data generation pipelines, SIGMA does not require any additional model training during synthesis. We use `Qwen2.5-Math-7B` for MCTS generation, a publicly available pretrained model from HuggingFace, requiring approximately 42 GPU hours on an RTX 4090 for generating full MCTS trees for a 15K-sample dataset.

For the refinement phase, we employ `GPT-4o-mini` to process the 15K MCTS trees, resulting in a total prompt token count of 33.6M and completion token count of 11.4M, with an API cost of 11.7 USD. Processing 60K examples with `GPT-4o-mini` costs only 47.6 USD. A detailed comparison of synthesis costs is shown in Table 6. Compared to DART-Math, which requires 3,840 GPU hours on A100s ($3,456 total), SIGMA reduces the total cost by more than **30**×. When replacing `GPT-4o-mini` with the open-weight `Qwen2.5-7B-Instruct` as the Critique and Revision model, the total cost further decreases to only $70, while maintaining comparable refinement quality. These results highlight SIGMA's efficiency and flexibility in reducing both computational and monetary overhead. In addition to computational efficiency, we analyze the token usage of SIGMA and sev-

Table 7: Token-level comparison across datasets. SIGMA achieves higher performance with a comparable or smaller token budget.

| Dataset | Tokens (LLaMA3-8B tokenizer) |
| --- | --- |
| DART-Math | 190,441,618 |
| MMIQC | 94,093,794 |
| RefAug | 54,228,136 |
| MetaMath | 110,278,040 |
| MathFusion | 40,437,234 |
| SIGMA | 44,064,782 |

eral baseline datasets to assess data efficiency. All token counts are measured using the LLaMA3-8B tokenizer. As shown in Table 7, SIGMA-15K contains 11.5M tokens, while SIGMA-60K contains 44M tokens—comparable to MathFusion's 60K dataset (40M tokens) yet achieving substantially higher accuracy. This demonstrates SIGMA's superior token-level data efficiency: with a similar token budget, it provides denser supervision and stronger generalization across benchmarks.

# E   Training Setup

All fine-tuning experiments were carried out on four NVIDIA H100 GPUs with DeepSpeed2 ZeRO [39] optimizations, operating in mixed precision [35] (FP16) to maximize memory efficiency and throughput.

The computation sequence token length was fixed at 4096 to capture long range mathematical reasoning. We set `per_device_train_batch_size=8` and used `gradient_accumulation_steps=4` to accumulate gradients.

We used the AdamW [25] optimizer with weight decay of 0.01. A cosine decay learning rate scheduler was applied with a linear warmup over the first 3 percent of total training steps (`warmup_ratio=0.03`) to prevent instability. All models were trained for 3 epochs.

Initial learning rates were tuned per model:

- DeepSeekMath-7B: $5 \times 10^{-5}$
- Mistral-7B: $4 \times 10^{-6}$
- LLaMA3-8B: $1 \times 10^{-5}$

We logged metrics at regular intervals during training and validation. Random seeds were fixed and model checkpoints were saved after each epoch to ensure reproducibility.

# F   Examples

In Appendix F, we present detailed examples and Q&A examples to provide deeper insight into the mechanism by which the SIGMA method refines reasoning data. Appendix F.1 illustrates how SIGMA refines reasoning steps on both geometry and algebra problems. Appendix F.2 presents additional SIGMA-refined reasoning paths for questions from MATH and GSM8K datasets.

## F.1   Detailed examples

---

### Example 1. Geometry

**Question:** The altitude to the hypotenuse of a triangle with angles of $30°$ and $60°$ is 3 units. What is the area of the triangle, in square units? Express your answer in simplest radical form.
#Picture below is compiled from Asymptote code in question line.

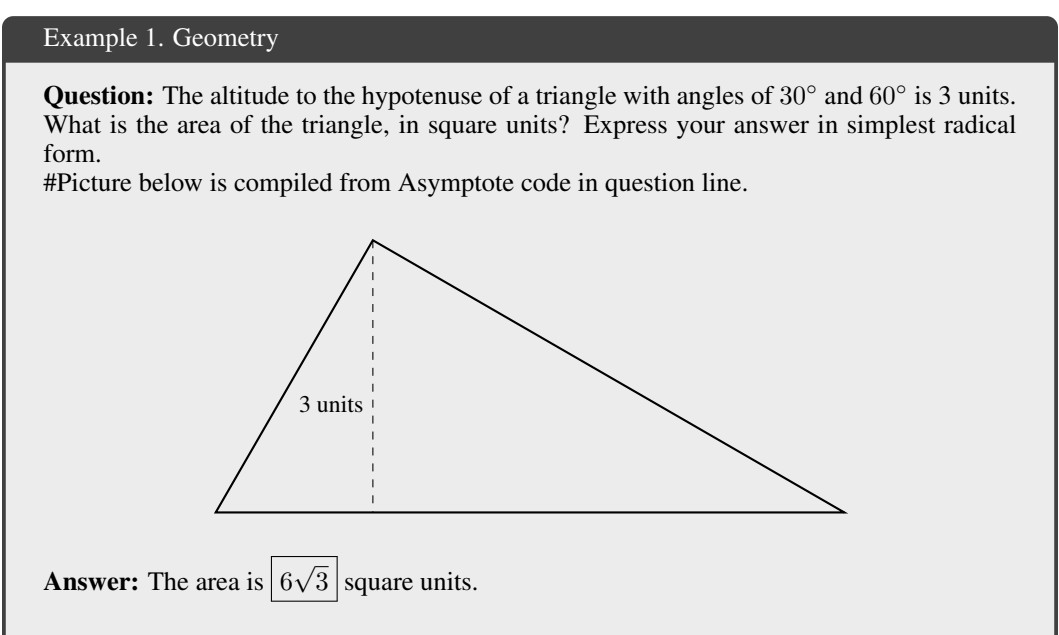

**Answer:** The area is $\boxed{6\sqrt{3}}$ square units.

---

**MCTS Selected Path:**

**(0-1):** Step: Calculate the length of the hypotenuse:

$$\text{hypotenuse} = 2 \cdot 3$$

**(0-1-2):** Step: Calculate the length of the base:

$$\text{base} = \frac{\text{hypotenuse}}{2}$$

**(0-1-2-1):** Step: Calculate the length of the altitude:

$$\text{altitude} = 3$$

**(0-1-2-1-1):** Step: Calculate the area of the triangle:

$$\text{area} = 0.5 \cdot \text{base} \cdot \text{altitude}$$

**The answer is:** $\boxed{4.5}$

**Step-by-Step Sibling Comparison and Critique:**

**Step 1:**

- **Sibling(0-2):** `Step:Calculate the length of the hypotenuse using the altitude and the 60-degree angle.We use the relationship in a` $30°$-$60°$-$90°$ `triangle.Since the altitude from the right angle to the hypotenuse is 3 units, the hypotenuse is:` `hypotenuse` $= 2\sqrt{3}$.

- **Sibling(0-0):** `Step:Calculate the length of the legs of the triangle.Using triangle ratios again, the longer leg (opposite the` $60°$ `angle) is:leg length` $= 3\sqrt{3}$.

- **Evaluation:** Step 0-0 incorrectly assumes a leg length without relating it to the given altitude; step 0-2 correctly uses the $30°$–$60°$–$90°$ property to link the altitude and hypotenuse; step 0-1 computed hypotenuse but resulted in wrong value.

**Step 2:**

- **Sibling(0-1-0):** `Step:Calculate the length of the side opposite the 30-degree angle:side_30 = hypotenuse / 2`

- **Sibling(0-1-1):** `Step:Compute the base using the Pythagorean theorem:base` $= \sqrt{\text{hypotenuse}^2 - \text{altitude}^2}$

- **Evaluation:** Step 0-1-0 conflates the $30°$ as opposite side with the base without justification; step 0-1-1 applies Pythagorean theorem correctly but adds unnecessary complexity; step 0-1-2 halves the hypotenuse that only holds in an isosceles right triangle.

**Step 3:**

- **Sibling(0-1-2-0):** `Step:Calculate the length of the height:height=3`

- **Sibling(0-1-2-2):** `Step:Compute the altitude using triangle legs:altitude=(leg1 * leg2) / hypotenuse`

- **Evaluation:** Step 0-1-2-0 merely restates the given;step 0-1-2-1 correctly extracts the altitude;step 0-1-2-2 applies a geometric formula but assumes leg values.

**Step 4:**

- **Sibling(0-1-2-1-0):** `Step:Calculate square:  base * altitude / 2 = 6`

- **Sibling(0-1-2-1-2):** `Step:Calculate answer:  0.5 * base * altitude = 6`

- **Evaluation:** Step 0-1-2-1-0 calculate the square; step 0-1-2-1-1 calculate the area of the triangle; step 0-1-2-1-2 compute the answer.

**Key points:**

- For step 0-1, explicitly derive the hypotenuse length from the altitude to the hypotenuse using the correct geometric relationship in a 30-60-90 triangle.
- For step 0-1-2, form and solve the altitude equation for that parameter.
- For step 0-1-2-1, substitute back to obtain all side lengths.
- For step 0-1-2-1-1, keep the square computing logic still.

**Optimzed Output:**
**Step 1:** Assign variables to the sides of the 30-60-90 triangle.
Let the shortest leg (opposite 30°) be $x$. Then the hypotenuse is $2x$, and the longer leg (opposite 60°) is $x\sqrt{3}$. Thus, the legs are $x$ and $x\sqrt{3}$, and the hypotenuse is $2x$.
**Step 2:** Use the given altitude length to solve for $x$.
We are told the altitude to the hypotenuse is 3 units, so:

$$3 = \frac{x\sqrt{3}}{2} \implies x = \frac{3 \times 2}{\sqrt{3}} = \frac{6}{\sqrt{3}}.$$

Rationalizing the denominator:

$$x = \frac{6}{\sqrt{3}} \times \frac{\sqrt{3}}{\sqrt{3}} = \frac{6\sqrt{3}}{3} = 2\sqrt{3}.$$

**Step 3:** Calculate the lengths of all sides using $x = 2\sqrt{3}$.
- Short leg (opposite 30°): $x = 2\sqrt{3}$.
- Longer leg (opposite 60°): $x\sqrt{3} = 2\sqrt{3} \times \sqrt{3} = 2 \times 3 = 6$.
- Hypotenuse: $2x = 2 \times 2\sqrt{3} = 4\sqrt{3}$.
**Step 4:** Calculate the area of the triangle.
The area of a right triangle is half the product of its legs:

$$\text{Area} = \frac{1}{2} \times (\text{short leg}) \times (\text{long leg}) = \frac{1}{2} \times 2\sqrt{3} \times 6 = \frac{1}{2} \times 12\sqrt{3} = 6\sqrt{3}.$$

The answer is: $\boxed{(6\sqrt{3})}$

---

### Example 2. Algebra

**Question:** Four positive integers $A$, $B$, $C$ and $D$ have a sum of 36. If $A + 2 = B - 2 = C \times 2 = D \div 2$, what is the value of the product $A \times B \times C \times D$?

**Answer:** The correct product is $\boxed{3840}$.

**MCTS Selected Path:**
**(0-2):** Step: Define the variables $A, B, C,$ and $D$.

**(0-2-2):** Step: Write the system of equations:

$$\begin{cases} A + B + C + D = 36, \\ A + 2 = B - 2, \\ B = C, \\ C = \dfrac{D}{2}. \end{cases}$$

**(0-2-2-1):** Step: Solve the system to obtain

$$A = 8, \quad B = 8, \quad C = 8, \quad D = 16.$$

**(0-2-2-1-0):** Step: Compute the product:
$$A \times B \times C \times D = 8 \times 8 \times 8 \times 16 = 4096.$$

The answer is: $\boxed{4096}$ (Incorrect)

**Step-by-Step Sibling Comparison and Critique:**
**Step 1:**
- **Sibling(0-0):** `Step:Define variables:A + B + C + D = 36`
- **Sibling(0-1):** `Step:Define sum:SUM = 36`
- **Evaluation:** Step 0-1 correctly begins by defining the total sum constraint explicitly; step 0-1 gives clear definition of total sum; step 0-2 omits in this step.

**Step 2:**
- **Sibling(0-2-0):** `Step:Define equations:A = C * 2 - 2; B = C * 2 + 2; D = C * 2 * 2`
- **Sibling(0-2-1):** `Step:Define relationships:A = B - 4; B = C * 2 + 2; D = A * 2 + 4`
- **Evaluation:** Step 0-2-0 expresses all variables in terms of a single variable $C$, which provides a clearer structure; step 0-2-1 give recurrent definition of relative variables;step 0-2-2 instead defines relationships in isolated pairs.

**Step 3:**
- **Sibling(0-2-2-0):** `Step:Solve equations:A + B + C + D = 36 ; A + 2 = B - 2 ; C * 2 = D / 2`
- **Sibling(0-2-2-2):** `Step:Solve equations:5 * C + 4 = 36`
- **Evaluation:** Step 0-2-2-0 captures the equality logic via $C * 2 = D/2$ correctly; step 0-2-2-2 express all variables using C; step 0-2-2-1 misrepresents this logic using $B = C$ and $C = D/2$.

**Step 4:**
- **Sibling(0-2-2-1-1):** `Step:Calculate requirement:  A * B * C * D = 4096`
- **Sibling(0-2-2-1-2):** `Step:Calculate answer:  A * B * C * D = 4096`
- **Evaluation:** Step 0-2-2-1-1 calculate the required output; step 0-2-2-1-2 calculate the answer ; step 0-2-2-1-0 compute the product.

**Key Points:**
- For step 0-2, explicitly state constraints in initial setup.
- For step 0-2-2, express all variables using a single reference variable.
- For step 0-2-2-1, correctly encode and solve the system with logical relationships.
- For step 0-2-2-1-0, keep the computing logic.

**Optimized Output:**
**Step 1:** Introduce a common value $k$ so that
$$A + 2 \;=\; B - 2 \;=\; 2C \;=\; \frac{D}{2} \;=\; k \quad \text{and} \quad A + B + C + D = 36.$$

**Step 2:** Express each variable in terms of $k$:
$$A = k - 2, \quad B = k + 2, \quad C = \frac{k}{2}, \quad D = 2k,$$

then substitute into the sum:
$$(k - 2) + (k + 2) + \tfrac{k}{2} + 2k = 36 \implies \frac{9k}{2} = 36 \implies k = 8.$$

**Step 3:** Determine the four integers:

$$A = 6, \quad B = 10, \quad C = 4, \quad D = 16,$$

which indeed satisfy both the sum and the four-way equality.

**Step 4:** Compute the product:

$$A \times B \times C \times D = 6 \times 10 \times 4 \times 16 = 3840.$$

The answer is: $\boxed{3840}$

## F.2 Q&A examples

### Example 1

**Question:** Mari made 4 more than five times as many decorated buttons as Kendra. Sue made half as many as Kendra. Sue made 6 buttons. How many did Mari make?

**Answer:**

**Step 1:** Sue made 6 buttons, so Kendra made $2 \times 6 = 12$.

**Step 2:** Mari made 4 more than $5\times$ Kendra's count: $5 \times 12 + 4$.

**Step 3:** Compute $5 \times 12 = 60$, then add 4 to get 64.

**Step 4:** Therefore, Mari made 64 buttons.

The answer is: $\boxed{64}$

### Example 2

**Question:** When three standard dice are tossed, the numbers $a, b, c$ are obtained. Find the probability that

$$(a - 1)(b - 1)(c - 1) \neq 0$$

**Answer:**

**Step 1:** Total outcomes $= 6^3 = 216$.

**Step 2:** $(a - 1)(b - 1)(c - 1) = 0$ iff at least one die shows 1.

**Step 3:** Favorable: none is $1 \implies 5^3 = 125$ outcomes.

**Step 4:** Probability $= \frac{125}{216}$.

**Step 5:** Hence the probability is $\frac{125}{216}$.

The answer is: $\boxed{\dfrac{125}{216}}$

### Example 3

**Question:** What is the minimum value of $y$ if $y = 3x^2 + 6x + 9$?

**Answer:**

**Step 1:** Rewrite or note vertex of parabola $3(x^2 + 2x) + 9 = 3(x + 1)^2 + 6$.

**Step 2:** Since $(x + 1)^2 \geq 0$, minimum occurs at $x = -1$.

**Step 3:** Substitute $x = -1$: $y = 3 \cdot 0 + 6 = 6$.

**Step 4:** Therefore, the minimum value is 6.

The answer is: $\boxed{6}$

### Example 4

**Question:** You are given a sequence of 58 terms $P + n$ where $P$ is the product of all primes $\leq 61$ and $n = 2, 3, \ldots, 59$. Let $N$ be the number of primes in this sequence. Find $N$.

**Answer:**
**Step 1:** If $n$ is prime (17 values), $P + n \equiv 0 \pmod{n}$ so composite.
**Step 2:** Remaining $58 - 17 = 41$ terms with composite $n$ may be prime.
**Step 3:** Known result for Euclid-type numbers gives exactly 16 primes among them.
**Step 4:** Thus $N = 16$.
The answer is: $\boxed{16}$

## Example 5

**Question:** A rectangle has length 40 m and width 20 m. A similar smaller rectangle has half the length and width. What is the area of the smaller rectangle?
**Answer:**
**Step 1:** Scale factor $= \frac{1}{2}$, so smaller sides are 20 m and 10 m.
**Step 2:** Area $= 20 \times 10 = 200 \text{ m}^2$.
**Step 3:** Alternatively area factor $(\frac{1}{2})^2 = \frac{1}{4}$ of 800 gives 200.
**Step 4:** Therefore, the area is 200.
The answer is: $\boxed{200}$

## Example 6

**Question:** Last year, Isabella took 7 math tests with distinct integer scores 91–100. Each running average after $n$ tests was an integer. She scored 95 on the 7th test. What was her 6th test score?
**Answer:**
**Step 1:** Let $S_n$ be sum after $n$ tests; $S_n \equiv 0 \pmod{n}$.
**Step 2:** From $S_7 = S_6 + 95 \equiv 0 \pmod 7 \implies S_6 \equiv 3 \pmod 7$.
**Step 3:** Also $S_6 \equiv 0 \pmod 6$, so $S_6 \equiv 24 \pmod{42}$.
**Step 4:** In allowable range $546 \le S_6 \le 600$, the valid sum is 570, giving $s_6 = 570 - S_5 = 100$.
**Step 5:** Therefore, the 6th score is 100.
The answer is: $\boxed{100}$

## Example 7

**Question:** Let $p, q$ be distinct solutions to $\dfrac{4x - 12}{x^2 + 2x - 15} = x + 2$. If $p > q$, find $p - q$.
**Answer:**
**Step 1:** Domain excludes $x = -5, 3$. Multiply by $x^2 + 2x - 15$ to get $x^3 + 4x^2 - 15x - 18 = 0$.
**Step 2:** Rational root $x = 3$ is extraneous. Factor gives $(x - 3)(x^2 + 7x + 6) = 0$.
**Step 3:** Solve $x^2 + 7x + 6 = 0 \implies x = -1, -6$.
**Step 4:** Thus $p = -1$, $q = -6$, so $p - q = 5$.
**Step 5:** Therefore, the difference is 5.
The answer is: $\boxed{5}$

## Example 8

**Question:** A vampire needs 7 gallons of blood per week, sucking 2 pints per person. How many people per day must he feed on?
**Answer:**
**Step 1:** 7 gal $\times$ 8 pt/gal $= 56$ pints/week.
**Step 2:** At 2 pt/person, he needs $56/2 = 28$ people/week.
**Step 3:** Dividing by 7 days gives $28/7 = 4$ people/day.
**Step 4:** Therefore, he must feed on 4 people each day.

The answer is: $\boxed{4}$

---

## Example 9

**Question:** Marla is mixing lilac paint 70% blue, 20% red, rest white. If she adds 140 oz of blue, how many ounces of white does she add?
**Answer:**
**Step 1:** White%=100–(70+20)=10%.
**Step 2:** Total $T$: $0.7T = 140 \implies T = 200$ oz.
**Step 3:** White amount $= 0.1 \times 200 = 20$ oz.
**Step 4:** Therefore, she adds 20 ounces of white.
The answer is: $\boxed{20}$

---

## Example 10

**Question:** By partial fractions,

$$\frac{1}{x(x+1)(x+2)(x+3)(x+4)} = \frac{A}{x} + \frac{B}{x+1} + \frac{C}{x+2} + \frac{D}{x+3} + \frac{E}{x+4}$$

for some constants $A, B, C, D, E$. Find $A + B + C + D + E$.
**Answer:**
**Step 1:** Multiply both sides by $x(x+1)(x+2)(x+3)(x+4)$ to get

$$\begin{aligned}
1 = {} & A(x+1)(x+2)(x+3)(x+4) \\
& + B\,x(x+2)(x+3)(x+4) \\
& + C\,x(x+1)(x+3)(x+4) \\
& + D\,x(x+1)(x+2)(x+4) \\
& + E\,x(x+1)(x+2)(x+3).
\end{aligned}$$

**Step 2:** Evaluate at the roots:

$$\begin{aligned}
x = 0: \quad & 1 = 24A \implies A = \tfrac{1}{24}, \\
x = -1: \quad & 1 = -6B \implies B = -\tfrac{1}{6}, \\
x = -2: \quad & 1 = 4C \implies C = \tfrac{1}{4}, \\
x = -3: \quad & 1 = -6D \implies D = -\tfrac{1}{6}, \\
x = -4: \quad & 1 = 24E \implies E = \tfrac{1}{24}.
\end{aligned}$$

**Step 3:** Sum the constants:

$$A + B + C + D + E = \tfrac{1}{24} - \tfrac{4}{24} + \tfrac{6}{24} - \tfrac{4}{24} + \tfrac{1}{24} = \tfrac{1-4+6-4+1}{24} = \tfrac{0}{24} = 0.$$

**Step 4:** Therefore, the sum is zero.
The answer is: $\boxed{0}$

---

## Example 11

**Question:** Let $P$ be a point outside of circle $O$. A segment is drawn from $P$ such that it is tangent to circle $O$ at $T$. Meanwhile, a secant from $P$ intersects $O$ at $A$ and $B$, with $PA < PB$. If $PA = 3$ and $PT = AB - PA$, then what is $PB$?
**Answer:**
**Step 1:** By the Power of a Point, $PT^2 = PA \cdot PB$.
**Step 2:** Since $AB = PB - PA$, the condition $PT = AB - PA$ gives $PT = (PB - PA) -$

$PA = PB - 2PA$, so with $PA = 3$ we have $PT = PB - 6$.

**Step 3:** Substitute into the power relation: $(PB - 6)^2 = 3\,PB$.

**Step 4:** Expand: $PB^2 - 12PB + 36 = 3PB$, hence $PB^2 - 15PB + 36 = 0$.

**Step 5:** Solve via the quadratic formula:

$$PB = \frac{15 \pm \sqrt{225 - 144}}{2} = \frac{15 \pm 9}{2},$$

yielding $PB = 12$ or $PB = 3$.

**Step 6:** Discard $PB = 3$ since $PB > PA = 3$, so $PB = 12$.

The answer is: $\boxed{12}$

