# OpenReview forum: "SIGMA: Refining Large Language Model Reasoning via Sibling-Guided Monte Carlo Augmentation"
_NeurIPS.cc/2025/Conference — NeurIPS 2025 poster_

### Official Review · Reviewer_UihT · 2025-07-02

**Clarity:** 2
**Significance:** 4
**Originality:** 3
**Rating:** 5
**Confidence:** 4

**Summary:**

The authors propose SIGMA, a refined version of MCTS that uses an oracle LLM prompted with sibiling node reasoning traces to generate feedback to refine the selected reasoning path. SIGMA is used to generate low sample reasoning datasets for fine-tuning which is empirically validated with three model variants. The authors find that SIGMA outperforms other methods in terms of fine-tuning sample efficiency and is competitive with much larger datasets such as DART-math.

**Questions:**

### Key questions to improve my score:
* What is the synthesis cost of SIGMA vs. other baselines such as DART-Math or MathFusion? Is SIGMA more optimal in terms of number of samples vs. cost to generate than the studied baselines?
* What is the total number of completion tokens in SIGMA vs. the other baselines? When we analyze the main results in terms of accuracy vs. completion tokens, do the main findings hold? Ultimately, our fine-tuning fine-tuning cost will scale with total tokens, not samples.
* How does Qwen-2.5-Math-7B perform when fine-tuned on the SIGMA dataset? Does SIGMA enable self-distillation?
* Some outputs from $\mathcal{C}\_{LLM}$ do not actually appear to contain a critique. Eg., from the geometry question: “Evaluation: Step 0-1-2-1-0 calculate the square; step 0-1-2-1-1 calculate the area
of the triangle; step 0-1-2-1-2 compute the answer.”. $\mathcal{C}\_{LLM}$ appears to simply be describing the sibling node approaches but doesn’t offer any specific critiques. Could the authors speculate on whether the more robust critiques could be extracted from $\mathcal{C}_{LLM}$  and if so, how?

### Minor issues, suggestions, typos
* Exploring additional oracle models, candidate generation models, and tasks would make the work more impactful. Given that SIGMA produces natural language feedback, it may be good fit for tasks where traditional verifiers are not available (eg., creative writing).
* Consider my comments regarding the overall presentation above. I believe the work would benefit from improvements in this area.
* Typos:
	* L145: “To”
	* L148: “Subscript” -> “Superscript”
	* L159: “operats”

**Ethical Concerns:**

["NO or VERY MINOR ethics concerns only"]

**Final Justification:**

My original key concerns regarding the synthesis cost of SIGMA and the ability to apply the method to a self-distillation setting were adequately addressed by the rebuttal provided by the authors. The abation using different models for $\mathcal{C}\_{LLM}$ and $\mathcal{R}_{LLM}$ have also improved my confidence that the proposed method is generalizable.

**Limitations:**

More discussion on the synthesis cost and total number of tokens in SIGMA datasets is required to fully understand the impact and limitations of the work.

**Quality:**

3

**Strengths And Weaknesses:**

## Strengths
* The proposed method is intuitive and well motivated.
* It appears the use of sibling guidance in MCTS is novel in the studied setting.
* A diverse set of models and math datasets are empirically studied.
* The result is significant and represents a valuable contribution to an important area of study. Improving sample efficiency is a long-standing challenge of the field.
* The proposed method is automated and scalable, potentially enabling the curation of large, high quality datasets.

## Weaknesses
* My primary concerns regarding this work are related to its motivation, some absent analysis, and presentation.
* The synthesis cost of SIGMA is not discussed. While I am convinced that SIGMA can be used to generate high-quality training samples, it’s unclear what the cost is to generate the dataset. This information would help motivate the selection of using SIGMA to generate a bespoke dataset versus using a larger dataset such as DART-math.
* The total tokens in the SIGMA dataset is unclear. Since we backpropagate across all completion tokens during fine-tuning, an important but absent discussion is the total number of tokens across SIGMA and baseline datasets.
* Only GPT-4o-mini is explored as $\mathcal{C}\_{LLM}$ and $\mathcal{R}_{LLM}$.
* Qwen-2.5-Math-7B is used as the candidate generator to construct search trees. One particularly interesting analysis, which was not included, is to fine-tune the same model on the SIGMA dataset to observe if SIGMA can be effectively leveraged in the self-distillation setting.
* Presentation concerns:
	* Authors should cite sections of the appendix in the main body where relevant. For example, when discussing training hyperparameters that are not explicitly stated in the main body.
	* The 60k SIGMA results presented in Appendix A are the most significant, I suggest moving these to the main body in lieu of Table 1. Highlighting that at 60k SIGMA enables higher quality models than even DART-Math or models trained on the full MathFusionQA dataset would improve the impact of the work.
	* Overall I find the formalization of Textual Gradient Descent to be more confusing than illustrative. Equation 5 can simply be re-written as $\tilde{T}\_p = \mathcal{R}\_{LLM}(T_p, \mathcal{C}_{LLM}(T_p | T_s \in \mathcal{S}_p) )$ without requiring us to stretch the definitions of terms such as gradient, loss, and optimizer to contexts beyond which they normally apply in ML literature. I understand that the authors are following the lead of [43] here; however, I believe the method could be more plainly stated without losing meaning and would improve the overall clarity.

---

> ### Author Rebuttal · Authors · 2025-07-31
>
> We appreciate the reviewer for the thoughtful feedback，recognizing (i) The proposed method is intuitive and well motivated, automated and scalable (ii) contribution in improving sample efficiency.We've addressed your concerns below.
> >The synthesis cost of SIGMA is not discussed...
>
> We appreciate the reviewer’s concern regarding the synthesis cost of SIGMA. As a data-efficient method, maintaining low synthesis cost and computation cost is central to its practicality and effectiveness.
>
> SIGMA does not require any additional model training during the data synthesis phase. We use Qwen-2.5-Math-7B for original MCTS generation,which is a publicly available pretrained models from HuggingFace. Following the MCTS data generation procedure of AlphaMath [6], using Qwen-2.5-Math-7B to generate full MCTS trees for a 15K dataset requires ~42 GPU hours on RTX 4090.
>
> For the refinement phase, we use GPT-4o-mini to process the 15K MCTS trees.The total prompt token count is 33.6M and the total completion token count is 11.42M, resulting in a total cost of 11.7 USD.Processing 60K MCTS on GPT-4o-mini costs 47.6 USD.Detailed comparison for synthesis cost are listed below. The GPU price data are from vast​.​ai.
>
> | Method | GPU Type | GPU Hours | Price per hour(USD)  | API Cost（USD） | Total Price(USD) |
> |:---:|:---:|:---:|:---:|:---:|:---:|
> | DART-Math | A100 | 3840 | 0.9 | – | 3456 |
> | SIGMA(GPT4o-mini as Crtique and Revision model) | RTX 4090 | 168 | 0.35 | 47.6 | 106.4 |
> | SIGMA(Qwen2.5-7B-Instructas as Crtique and Revision model) | RTX 4090 | 200 | 0.35 | – | 70 |
>
> Compared to alternative data generation pipelines, SIGMA offers a competitive balance of low computational cost, high data efficiency, and improved overall accuracy. We will include detailed computational cost analysis in the camera-ready version.Moreover, thanks to the flexibility of the SIGMA framework, we can also substitute the proprietary GPT‑4o‑mini with an open‑source 7B model, reducing costs by over 30% while incurring only a 4.6% drop in performance.
>
> >The total tokens in the SIGMA dataset is unclear...
>
> We appreciate the reviewer’s detailed concern for computational cost for training that dives into token-level.
>
> To support the claim of SIGMA’s data efficiency, we report the total number of tokens used in SIGMA-15K, SIGMA-60K, and other baseline datasets, all measured using the LLaMA3-8B tokenizer. SIGMA-15K contains 11.5M tokens,It is worth noting that, with a comparable number of tokens per question, DeepSeek-7B-SIGMA (15K) outperforms MathFusion trained on 30K examples. while SIGMA-60K contains 44M tokens, which is roughly comparable to the token count of MathFusion’s 60K dataset, SIGMA achieves state-of-the-art performance, highlighting its efficiency in data usage.
>
> | Dataset | Tokens (Llama3-8B tokenizer) |
> |:---:|:---:|
> | DART-Math | 190441618 |
> | MMIQC  | 940937974 |
> | RefAug  | 54228136 |
> | MetaMath | 110278040 |
> | MathFusion | 40437234 |
> | SIGMA | 44064782 |
>
> Moreover, the low cost of data synthesis in SIGMA makes it scalable to larger datasets without prohibitive resource demands. We will include detailed token statistics in the camera-ready version.
> >Only GPT-4o-mini is explored as $C_{LLM}$ and $R_{LLM}$.
>
> Thank you for the insightful question. For the $C_{LLM}$ and $R_{LLM}$, GPT-4o-mini was chosen as a cost-effective option. However, our method does not fully rely on closed-source models. To support this, we have included additional results using Qwen2.5-72B-Instruct, and Qwen2.5-Math-7B-Instruct as $C_{LLM}$ and $R_{LLM}$ to synthesise 3 new 15K datasets, and SFT on models in our main experiment.Switching $C_{LLM}$ and $R_{LLM}$ to weaker open-source LLMs resultd in a comparable performance compared with closed-source model as $C_{LLM}$ and $R_{LLM}$.Under Qwen-2.5-72B-instruct，the results achieves 34.3 on Llama3-8B，which is still comparable；31 on mistral，and 46.5 on deepseekMath-7B,which remains state-of-the-art with other methods.
>
> | **15K Samples** | MATH | GSM8K | College | DM | Olympiad | Theorem | AVG |
> |:---:|:---:|:---:|:---:|:---:|:---:|:---:|:---:|
> | **Llama3-8B Series** |  |  |  |  |  |  |  |
> | Qwen2.5-7B-Instruct as Critique and Revision Model | 35.3  | 77.1  | 21.7  | 39.5  | 9.6  | 16.9  | 33.3  |
> | Qwen2.5-72B-Instruct as Critique and Revision Model | 36.3  | 78.9  | 23.2  | 39.4  | 9.8  | 18.0  | 34.3  |
> | GPT4o-mini as Critique and Revision Model | 36 | 82 | 24.2 | 42 | 10.5 | 22 | 36.1 |
> | MathFusion（30K Average Score） | 37.2  | 76.7  | 23.9  | 37.1  | 12.1  | 17.1  | 34.1  |
> | **Mistral-7B-v0.1 Series** |  |  |  |  |  |  |  |
> | Qwen2.5-7B-Instruct as Critique and Revision Model | 28.8 | 79.3 | 18.4 | 33.3 | 6.7 | 13.5 | 30 |
> | Qwen2.5-72B-Instruct as Critique and Revision Model | 30.6 | 76.8 | 21.1 | 35.3 | 6.8 | 15.6 | 31 |
> | GPT4o-mini as Critique and Revision Model | 30 | 75.3 | 20.8 | 39.5 | 7.7 | 16.3 | 31.6 |
> | MathFusion（30K Average Score） | 30.0  | 74.0  | 18.5  | 25.7  | 9.2  | 14.5  | 28.6  |
> | **DeepSeekMath-7B-Base Series** |  |  |  |  |  |  |  |
> | Qwen2.5-7B-Instruct as Critique and Revision Model | 49.8  | 80.6  | 38.3  | 60.6  | 18.4  | 24.1  | 45.3  |
> | Qwen2.5-72B-Instruct as Critique and Revision Model | 51.3  | 81.9  | 39.9  | 64.8  | 17.5  | 23.8  | 46.5  |
> | GPT4o-mini as Critique and Revision Model | 52.2 | 81.1 | 37.4 | 64.5 | 20.3 | 26.1 | 47 |
> | MathFusion（30K Average Score） | 49.8  | 76.0  | 38.2  | 60.7  | 20.0  | 21.9  | 44.4  |
> | **Qwen2.5-Math-7B Series** |  |  |  |  |  |  |  |
> | Qwen2.5-7B-Instruct as Critique and Revision Model | 74.0  | 89.9  | 48.5  | 83.1  | 39.3  | 40.6  | 62.6  |
> | Qwen2.5-72B-Instruct as Critique and Revision Model | 76.9  | 91.2  | 48.6  | 87.0  | 39.6  | 39.0  | 63.7  |
> | GPT4o-mini as Critique and Revision Model | 79.9  | 89.2  | 45.2  | 88.8  | 43.4  | 47.0  | 65.6  |
>
> We are deeply grateful for your insightful comments on self-distillation. Taking your suggestion to heart, we used Qwen2.5-7B-Instruct — a model with the same parameter size and comparable performance to Qwen2.5-Math-7B — as both the critique and revision model to synthesize 15K data points. To our delight, we found that despite Qwen2.5-7B-Instruct being less capable than GPT4o-mini, the fine-tuned model exhibited only a very minimal performance drop on this dataset (from 65.6 to 62.6).
>
> >Qwen-2.5-Math-7B is used as the candidate generator to construct search trees. One particularly interesting analysis, which was not included...
>
> Thank you for suggesting exlporation on self-distillation setting. In the main experiments, we did not finetune Qwen2.5-Math-7B because existing strong baselines such as DART and MathFusion and other baselines have not reported results on this model. To ensure fair and verifiable comparisons, we focused on base models with publicly available results.
>
> To further assess the self-distillation ability of our method, we additionally conducted finetuning experiments on Qwen2.5-Math-7B.SIGMA remains state-of-the-art performace on Qwen2.5-Math-7B, reaching 79.92 for MATH, 89.23 for GSM8K and 65.6 on all datasets average,surpassing other methods such as Eurus and OpenMath.The results are shown in the following table.
>
>
> | Qwen2.5-Math-7B (Fine-tuned) | # Samples | MATH | GSM8K | College | DM | Olympiad | Theorem | AVG |
> |:---:|:---:|:---:|:---:|:---:|:---:|:---:|:---:|:---:|
> | Rstar@Greedy | 747K | 78.40  | 89.70  | **52.50** | – | **47.10** | – | – |
> | Eurus-2-7B-PRIME | 230K | 53.88  | 42.99  | 20.97  | 47.30  | 22.22  | 18.62  | 34.33  |
> | OpenMath-Nemotron-7B | 3.2M | 40.52  | 73.39  | 32.93  | 54.70  | 11.41  | 13.63  | 37.76  |
> | qwen2.5math-instruct | 2.5M | 82.16  | 95.15  | 46.80  | 82.00  | 41.60  | 40.75  | 64.74  |
> | AceMath-7B-Instruct | 2.3M | 75.40  | 58.38  | 45.82  | 59.70  | 39.85  | 30.75  | 51.65  |
> | DART-Math-60K | 60K | 63.14  | 91.28  | 47.23  | 74.90  | 26.81  | 35.38  | 56.46  |
> | MathFusion-60K | 60K | 56.14  | 59.59  | 37.08  | 67.70  | 41.19  | 27.87  | 48.26  |
> | SIGMA-60K | 60K | 79.92  | 89.23  | 45.17  | **88.80** | 43.41  | **47.00** | **65.59** |
>
> >Some outputs from $C_{LLM}$ do not actually appear to contain a critique.$C_{LLM}$ appears to simply be describing the sibling node approaches but doesn’t offer any specific critiques. Could the authors speculate on whether the more robust critiques could be extracted from $C_{LLM}$ and if so, how?
>
> This is an important question and directly relates to the effectiveness of the SIGMA method.
>
> In most mathematical problems, there exists a distinction between critical step（logical steps that directly determine correctness) and non-critical steps(routine computations or definition lookups). To ensure that all potentially critical reasoning steps are considered, SIGMA applies step-level refinement throughout the entire MCTS trajectory. This full-chain processing inevitably includes some non-critical steps that may not benefit significantly from refinement. Occasionally, LLMs may fail to generate meaningful critiques for these simpler steps.That's why even if the refinement of simple steps may not directly improve final accuracy, including them is necessary to ensure robustness across the entire reasoning paths.
>
> The processing of entire reasoning paths does not compromise the overall effectiveness of SIGMA. Because our refinement operates at the step level, and each selected step $T_p$ shares the same parent as its sibling steps $T_s$, any critical step will be refined using other steps that are equally critical. This guarantees that when refinement matters most, the available sibling signals are relevant and high quality.
>
>  We will clarify this point in the camera-ready version.
>
> **Final Notes:** We appreciate for recognizing the motivation and efficiency of our method. We've added clarifications on synthesis cost, token usage, and open-source/self-distillation results. We hope these updates address your concerns and highlight SIGMA's practicality.

---

> > ### Comment · Reviewer_UihT · 2025-08-03
> >
> > I thank the authors for their detailed rebuttal. In particular, I find the new data provided w.r.t. sampling efficiency at the token level and the self-distillation results particularly convincing that SIGMA represents a valuable contribution to the proceedings. As such, I have increased my original score to accept.

---

> ### Author Response · Authors · 2025-08-04
>
> Dear Reviewer UihT,
>
> Thank you for your thoughtful and supportive feedback. Your suggestions were very helpful in improving the quality and clarity of our work, and they helped us better appreciate the potential of SIGMA. We will incorporate the additional results and corresponding revisions into the final version of the paper. We truly appreciate your time and effort in reviewing our work.
>
> Best regards,
> Authors

---

### Official Review · Reviewer_Jhvz · 2025-07-02

**Clarity:** 3
**Significance:** 3
**Originality:** 3
**Rating:** 5
**Confidence:** 4

**Summary:**

The paper presents a framework that reuses the discarded alternative generations (sibling nodes) produced during LLM test-time search via Monte Carlo Tree Search (MCTS). These sibling nodes were discarded in prior work [1]. The framework instead leverages a critique LLM (GPT-4) to provide feedback on these sibling nodes, which is then used by a revision LLM (also GPT-4) to rewrite the generation step into a chain of thought, thereby improving the quality of supervised training data.
This process enables the authors to generate smaller but higher-quality training data compared to using MCTS alone. The approach is evaluated on three 7–8B parameter models across several math reasoning datasets and benchmarks, including Vanilla MCTS, Random-Sibling MCTS, Self-Refine, Tree-of-Thoughts-20, direct CoT SFT, and MathFusion-60K.

[1] Chen, G., Liao, M., Li, C., & Fan, K. (2024). AlphaMath Almost Zero: Process Supervision without Process (arXiv:2405.03553). arXiv. https://doi.org/10.48550/arXiv.2405.03553

**Questions:**

How would SIGMA’s performance degrade when you change the critique and revision LLMs to weaker open-source models?
Have you measured how the number of sibling nodes affects SIGMA's performance?

**Ethical Concerns:**

["NO or VERY MINOR ethics concerns only"]

**Final Justification:**

The authors provided many experiments underlying their claim, so I suggest accepting the paper.

**Quality:**

3

**Strengths And Weaknesses:**

Strenghts:

Test-time search in LLMs and self-refining reasoning LLMs are highly relevant topics, and this paper is well motivated and well written.

While I find the methodological contribution narrow as it mainly relies on using all generations and as such modest, the repurposing of the otherwise discarded siblings is carried out methodologically sound. The evaluation on different baselines (Vanilla MCTS, Random-Sibling MCTS, Self-Refine, Tree-of-Thoughts-20, direct CoT SFT, MathFusion-60 K) is comprehensive.

The incorporation of critique and refiner models is a simple approach to improve certain siblings and can be easily applied.

Weaknesses:

As Mentioned already in the strengths, the novelty lies in a narrow domain, using all generations for learning as an addition to previous work using MCTS. There is work in that direction I find not adressed in this submission, such as [2], that in a similar manner use generations for each token in a prompt for supervised training. While the authors add a trick with the critique- and refinement LLMs, this shows some overlap. As such it further limits novelty in my view.

While evaluation across benchmarks is comprehensive, I find evidence for the generalizability across models and domains could be more robust:
1. The approach is limited to the math domain, so I am unsure how well it generalizes to more open domains such as commonsense reasoning.
2. While it is evaluated across model families, the evaluation is limited to similar sizes and doesn't use specialized models, as [1] does.

---

> ### Author Rebuttal · Authors · 2025-07-31
>
> We appreciate the reviewer for the thoughtful feedback，recognizing (i) our methods are well motivated and can be easily applied (ii) experiments and evaluations are comprehensive. We've addressed your concerns below.
> >There is work in that direction I find not adressed in this submission, such as [2], that in a similar manner use generations for each token in a prompt for supervised training. While the authors add a trick with the critique- and refinement LLMs, this shows some overlap.
>
> We appreciate the reviewer for highlighting related work such as Self-RAG [2]. While both methods leverage partial generations for supervision, there are key differences that distinguish SIGMA.
>
> (i) Self-RAG evaluates token-wise fragments and discards suboptimal completions, selecting only one high-quality continuation. In contrast, SIGMA explicitly retains and reuses sibling answers that are typically discarded, using them as the basis for critique generation. This reuse of suboptimal trajectories is a central motivation of SIGMA and contributes to its data efficiency.
>
> (ii) In Self-RAG, the selected fragments are sequential and share a context-dependent relationship. In SIGMA, by contrast, the selected node $T_p$ and its sibling nodes $T_s$ are parallel completions generated from the same parent node during MCTS. This structural alignment ensures that the critique model can extract meaningful comparative signals from $T_s$ to revise $T_p$, rather than introducing unrelated information.
>
> We will include a discussion of this distinction in the camera-ready version.
>
> >The approach is limited to the math domain, so I am unsure how well it generalizes to more open domains such as commonsense reasoning.
>
> We appreciate the reviewer for the helpful suggestion.We agree that evaluating the generalizability of SIGMA beyond mathematical reasoning is important. Owing to its design that relies entirely on textual signals for refining reasoning paths, SIGMA holds strong potential in non-mathematical domains as well.
>
> Following the experimental setup of RevThink[1], we applied SIGMA to CommonsenseQA, StrategyQA, and ARC-Challenge. We conducted LoRA finetuning on Mistral-7B-Instruct-v0.3 and Gemma-7B-Instruct to assess SIGMA’s cross-domain generalization ability. SIGMA outperformed RevThink and other baselines, achieving state-of-the-art performance(75.74 AVG on Mistral-7B-Instruct-v0.3 and 73.63 AVG on Gemma-7B-Instruct) on these benchmarks.
>
> Detailed results are provided in the table below.
>
> | Methods | Mistral-7B-Instruct-v0.3 |  |  |  | Gemma-7B-Instruct |  |  |  |
> |:---:|:---:|:---:|:---:|:---:|:---:|:---:|:---:|:---:|
> |  | SQA | CSQA | ARC | AVG | SQA | CSQA | ARC | AVG |
> | Zero-shot | 53.89 | 62.57 | 73.68 | 63.38 | 56.33 | 66.26 | 68.34 | 63.64 |
> | SKD | 63.76 | 71.86 | 74.66 | 70.09 | 56.77 | 72.48 | 73.29 | 67.51 |
> | Distill Step-by-Step | 64.19 | 71.92 | 75.32 | 70.48 | 56.77 | 73.01 | 72.92 | 67.57 |
> | Rephrase Question | 65.07 | 70.19 | 74.51 | 69.92 | 54.15 | 70.22 | 72.37 | 65.58 |
> | Question Aug | 65.07 | 72.23 | 73.32 | 70.21 | 57.21 | 68.11 | 72.74 | 66.02 |
> | Answer Aug | 66.38 | 69.12 | 76.77 | 70.76 | 57.21 | 73.01 | 73.92 | 68.05 |
> | Revthink | 70.97 | 75.76 | 78.5 | 75.08 | 64.19 | 74.53 | 75.09 | 71.27 |
> | SIGMA(Ours) | 71.62 | 76.3 | 79.3 | 75.74 | 67.4 | 75.4 | 78.1 | 73.63 |
>
> We will include the discussion on generalization and the corresponding experimental results in the camera-ready version.
>
> >While it is evaluated across model families, the evaluation is limited to similar sizes and doesn't use specialized models, as [1] does.How would SIGMA’s performance degrade when you change the critique and revision LLMs to weaker open-source models? Have you measured how the number of sibling nodes affects SIGMA's performance?
>
> We appreciate the reviewer for the valuable suggestions regarding evaluation across model strengths and sibling node configurations.
>
> To further assess the self-improvement ability of our method, we additionally conducted finetuning experiments on Qwen2.5-Math-7B.SIGMA remains state-of-the-art performace on Qwen2.5-Math-7B, reaching 79.92 for MATH, 89.23 for GSM8K and 65.59 on all datasets average,surpassing other methods such as Eurus and OpenMath.The results are shown in the following table.
> | Qwen2.5-Math-7B (Fine-tuned) | # Samples | MATH | GSM8K | College | DM | Olympiad | Theorem | AVG |
> |:---:|:---:|:---:|:---:|:---:|:---:|:---:|:---:|:---:|
> | Rstar@Greedy | 747K | 78.40  | 89.70  | **52.50** | – | **47.10** | – | – |
> | Eurus-2-7B-PRIME | 230K | 53.88  | 42.99  | 20.97  | 47.30  | 22.22  | 18.62  | 34.33  |
> | OpenMath-Nemotron-7B | 3.2M | 40.52  | 73.39  | 32.93  | 54.70  | 11.41  | 13.63  | 37.76  |
> | qwen2.5math-instruct | 2.5M | 82.16  | 95.15  | 46.80  | 82.00  | 41.60  | 40.75  | 64.74  |
> | AceMath-7B-Instruct | 2.3M | 75.40  | 58.38  | 45.82  | 59.70  | 39.85  | 30.75  | 51.65  |
> | DART-Math-60K | 60K | 63.14  | 91.28  | 47.23  | 74.90  | 26.81  | 35.38  | 56.46  |
> | MathFusion-60K | 60K | 56.14  | 59.59  | 37.08  | 67.70  | 41.19  | 27.87  | 48.26  |
> | SIGMA-60K | 60K | 79.92  | 89.23  | 45.17  | **88.80** | 43.41  | **47.00** | **65.59** |
>
> For the critique and revision LLMs, GPT-4o-mini was chosen as a cost-effective option. However, our method does not depend on closed-source models. To support this, we have included additional results using, Qwen2.5-72B-Instruct, and Qwen2.5-Math-7B-Instruct as critique and revision LLMs to synthesise 3 new 15K datasets, and SFT on models in our main experiment.Switching refinement model to weaker open-source LLMs resultd in a comparable performance compared with closed-source model as refinment models.Under Qwen-2.5-72B-instruct，the results achieves 34.3 on Llama3-8B，which is still comparable；31 on mistral，and 46.5 on deepseekMath-7B,which remains state-of-the-art with other methods.
>
> | **15K Samples** | MATH | GSM8K | College | DM | Olympiad | Theorem | AVG |
> |:---:|:---:|:---:|:---:|:---:|:---:|:---:|:---:|
> | **Llama3-8B Series** |  |  |  |  |  |  |  |
> | Qwen2.5-7B-Instruct as Critique and Revision Model | 35.3  | 77.1  | 21.7  | 39.5  | 9.6  | 16.9  | 33.3  |
> | Qwen2.5-72B-Instruct as Critique and Revision Model | 36.3  | 78.9  | 23.2  | 39.4  | 9.8  | 18.0  | 34.3  |
> | GPT4o-mini as Critique and Revision Model | 36 | 82 | 24.2 | 42 | 10.5 | 22 | 36.1 |
> | MathFusion（30K Average Score） | 37.2  | 76.7  | 23.9  | 37.1  | 12.1  | 17.1  | 34.1  |
> | **Mistral-7B-v0.1 Series** |  |  |  |  |  |  |  |
> | Qwen2.5-7B-Instruct as Critique and Revision Model | 28.8 | 79.3 | 18.4 | 33.3 | 6.7 | 13.5 | 30 |
> | Qwen2.5-72B-Instruct as Critique and Revision Model | 30.6 | 76.8 | 21.1 | 35.3 | 6.8 | 15.6 | 31 |
> | GPT4o-mini as Critique and Revision Model | 30 | 75.3 | 20.8 | 39.5 | 7.7 | 16.3 | 31.6 |
> | MathFusion（30K Average Score） | 30.0  | 74.0  | 18.5  | 25.7  | 9.2  | 14.5  | 28.6  |
> | **DeepSeekMath-7B-Base Series** |  |  |  |  |  |  |  |
> | Qwen2.5-7B-Instruct as Critique and Revision Model | 49.8  | 80.6  | 38.3  | 60.6  | 18.4  | 24.1  | 45.3  |
> | Qwen2.5-72B-Instruct as Critique and Revision Model | 51.3  | 81.9  | 39.9  | 64.8  | 17.5  | 23.8  | 46.5  |
> | GPT4o-mini as Critique and Revision Model | 52.2 | 81.1 | 37.4 | 64.5 | 20.3 | 26.1 | 47 |
> | MathFusion（30K Average Score） | 49.8  | 76.0  | 38.2  | 60.7  | 20.0  | 21.9  | 44.4  |
> | **Qwen2.5-Math-7B Series** |  |  |  |  |  |  |  |
> | Qwen2.5-7B-Instruct as Critique and Revision Model | 74.0  | 89.9  | 48.5  | 83.1  | 39.3  | 40.6  | 62.6  |
> | Qwen2.5-72B-Instruct as Critique and Revision Model | 76.9  | 91.2  | 48.6  | 87.0  | 39.6  | 39.0  | 63.7  |
> | GPT4o-mini as Critique and Revision Model | 79.9  | 89.2  | 45.2  | 88.8  | 43.4  | 47.0  | 65.6  |
>
> For the influence of the number of sibling nodes, We have conducted additional experiments to study the effect of varying the number of sibling nodes used during refinement.We choose DeepSeekMath-7B and Llama3-8B as Math-Specialized Base Model and General Base model.Lines with suffix MCTS are extracted from Table 2 in Section 4.4, page 9 of our paper.The results shows clearly monotonic increase trend when raising numbers of siblings(33.6->46.5->47.0 for DeepSeekMath and 23.6->35.0->36.1 for Llama3-8B)
>
> | Model(15K Samples) | MATH | GSM8K | College | DM | Olympiad | Theorem | AVG |
> |---|---|---|---|---|---|---|---|
> | Math-Specialized Base Model |  |  |  |  |  |  |  |
> | DeepSeekMath-7B-MCTS(0  Sibling) | 40.7 | 58.3 | 30.2 | 43.7 | 18.1 | 10.7 | 33.6 |
> | DeepSeekMath-7B(1 Sibling) | 50.7  | 81.3  | 35.1  | 65.5  | 19.6  | 26.9  | 46.5  |
> | DeepSeekMath-7B(2 Siblings) | 52.2  | 81.1  | 37.4  | 64.5  | 20.3  | 26.1  | 47.0  |
> | General Base Model |  |  |  |  |  |  |  |
> | Llama3-8B-MCTS(0  Sibling) | 27.8 | 40.2 | 18.3 | 34.3 | 12.6 | 8.5 | 23.6 |
> | Llama3-8B(1 Sibling) | 78.9  | 36.1  | 23.6  | 40.3  | 11.6  | 19.3  | 35.0  |
> | Llama3-8B(2 Siblings) | 36.0  | 82.0  | 24.2  | 42.0  | 10.5  | 22.0  | 36.1  |
>
> [1]Chen, J. C. Y., Wang, Z., Palangi, H., Han, R., Ebrahimi, S., Le, L., ... & Pfister, T. (2024). Reverse thinking makes llms stronger reasoners. arXiv preprint arXiv:2411.19865.
>
> **Final Note：** We thank the reviewer for recognizing the motivation and completeness of our method and experiments. We have added comparisons with Self-RAG, evaluated generalization to commonsense domains, and analyzed the effects of model strength and sibling count. We hope these additions fully address your concerns.

---

> > ### Comment · Reviewer_Jhvz · 2025-08-05
> > **Response to Authors**
> >
> > Thank you to the authors for the detailed response, addressing most of my concerns. I will raise my score that was leaning towards accept to accept.

---

> > > ### Author Response · Authors · 2025-08-05
> > >
> > > Dear Reviewer Jhvz,
> > >
> > > Thank you very much for your thoughtful and constructive feedback. We are glad that our response helped address most of your concerns, and we sincerely appreciate your decision to raise the score.
> > >
> > > Your comments were very helpful in guiding us to improve the clarity and presentation of our work. We will incorporate the corresponding revisions and improvements into the final version of the paper. Thank you again for your time, careful reading, and support throughout the review process.
> > >
> > > Best regards,
> > >
> > > Authors

---

### Official Review · Reviewer_5rCb · 2025-07-05

**Clarity:** 2
**Significance:** 2
**Originality:** 2
**Rating:** 3
**Confidence:** 4

**Summary:**

In this paper, the authors introduced a framework for refining the reasoning capabilities of large language models (LLMs) by leveraging sibling nodes in Monte Carlo Tree Search (MCTS), called SIGMA. SIGMA reintegrates discarded sibling nodes to enhance reasoning paths through a two-stage refinement process: a critique model discovers strengths and weaknesses across sibling sets, and a revision model refines the top-scoring trajectory based on the findings from the first step. The method improves reasoning trajectories, achieving state-of-the-art performance on the MATH benchmark with only 30K samples.

**Questions:**

The authors need to answer all the weak points listed above.

**Ethical Concerns:**

["NO or VERY MINOR ethics concerns only"]

**Limitations:**

Please check the weak points listed above.

**Paper Formatting Concerns:**

N.A.

**Quality:**

2

**Strengths And Weaknesses:**

1.The core idea of leveraging sibling information for refinement is not entirely novel. Similar concepts have been explored in other domains, such as symbolic supervision via LLM-generated feedback [43]. The paper does not provide a clear distinction or significant advancement over these existing methods.
2.The paper lacks a formal theoretical analysis of why the proposed method works and its potential limitations. A deeper understanding of the underlying mechanisms and potential failure modes would strengthen the paper
3.The performance of SIGMA is highly dependent on the quality of the sibling nodes generated during MCTS. If the sibling nodes are not well-formed or fail to capture robust reasoning behaviors, the transferred feedback may not provide significant benefits.
4. While SIGMA avoids full MCTS fine-tuning on large models, it still requires access to a reasonably strong small expert model trained with MCTS, which can be computationally expensive to obtain. The paper does not provide a detailed analysis of the trade-offs between the computational cost of training the small expert models and the benefits gained from using SIGMA.

---

> ### Author Rebuttal · Authors · 2025-07-31
>
> We appreciate the reviewer for the thoughtful feedback，We've addressed your concerns below.
> >1.The core idea of leveraging sibling information for refinement is not entirely novel. Similar concepts have been explored in other domains, such as symbolic supervision via LLM-generated feedback [43]. The paper does not provide a clear distinction or significant advancement over these existing methods.
>
> Our primary motivation is to improve data efficiency by reusing sibling trajectories from the MCTS process, which are typically discarded in prior work. TextGrad [43] serves as a key component in realizing this goal, but our contribution lies in how we apply it to exploit the structure of MCTS reasoning trees.
> Unlike previous methods that focus solely on the best trajectory, our approach systematically leverages feedback from suboptimal siblings to revise and improve selected steps. This enables more effective utilization of existing data, addressing the trade-off between accuracy and test-time scaling cost. Our experiments demonstrate that this strategy leads to consistent performance gains across multiple models and settings, validating its effectiveness.
> >2.The paper lacks a formal theoretical analysis of why the proposed method works and its potential limitations. A deeper understanding of the underlying mechanisms and potential failure modes would strengthen the paper
>
> We appreciate the reviewer’s comment regarding the need for a deeper understanding of SIGMA's underlying mechanisms and its potential limitations.
>
> SIGMA builds on the structural alignment of MCTS-generated reasoning trees, where each selected step $T_p$ and its sibling steps $T_s$ originate from the same parent and thus represent semantically parallel alternatives. This setting enables meaningful counterfactual supervision: by applying a critique model to $T_s$ and using its output to guide the revision of $T_p$, SIGMA transforms discarded reasoning attempts into constructive feedback. This aligns with principles of contrastive learning and error-based refinement, contributing to SIGMA’s high data efficiency.
>
> However, a potential failure mode arises when the input problems are too simple or formulaic, such that the resulting reasoning trajectories are dominated by non-critical steps (e.g., routine calculations or definition lookups). In such cases, SIGMA still performs step-level refinement uniformly, which may lead to unnecessary processing of low-impact steps. Moreover, language models may fail to produce substantive critiques for these trivial steps, making the refinement less effective and reducing the marginal utility per token. This can weaken SIGMA’s data-efficiency advantage in low-complexity settings.
>
> Nevertheless, because SIGMA refines each step individually and siblings share local context, critical steps are still improved using relevant sibling information. Thus, when refinement truly matters (i.e., on non-trivial reasoning segments), the method remains robust and effective.
>
> We will include this discussion of SIGMA’s underlying mechanism and its potential failure modes in the camera-ready version.
> >3.The performance of SIGMA is highly dependent on the quality of the sibling nodes generated during MCTS. If the sibling nodes are not well-formed or fail to capture robust reasoning behaviors, the transferred feedback may not provide significant benefits.
>
> The central contribution of our work lies in how to extract constructive revision signals for the selected step $T_p$ from its suboptimal siblings $T_s$. Our method does not refine the content of $T_p$ to be closer to the content of $T_s$. Instead, we first apply a critique model to analyze $T_s$, and then use a revision model to improve $T_p$ based on the critique outcomes.
>
> This design enables two key benefits:
> – Alternative reasoning paths in $T_s$ are leveraged to enrich the revision of $T_p$ without copying them directly.
> – Mistakes in $T_s$ are explicitly identified and used to detect and correct similar issues in $T_p$, rather than introducing the same errors into $T_p$.
>
> We will clarify these mechanisms in the camera-ready version.
>
> >4. While SIGMA avoids full MCTS fine-tuning on large models, it still requires access to a reasonably strong small expert model trained with MCTS, which can be computationally expensive to obtain. The paper does not provide a detailed analysis of the trade-offs between the computational cost of training the small expert models and the benefits gained from using SIGMA.
>
> We appreciate the reviewer’s concern regarding the computational cost of SIGMA. As a data-efficient method, maintaining low computation cost is central to its practicality and effectiveness.
>
> We clarify that SIGMA does not require any additional model training during the data synthesis phase. We use Qwen-2.5-Math-7B for original MCTS generation,which is a publicly available pretrained models from HuggingFace. Following the MCTS data generation procedure of AlphaMath [6], using Qwen-2.5-Math-7B to generate full MCTS trees for a 15K dataset requires ~42 GPU hours on RTX 4090.Generating MCTS for 60K dataset requires ~168 RTX 4009 GPU hours.
>
> For the refinement phase, we use GPT-4o-mini to process the 15K MCTS trees.The total prompt token count is 33.6M and the total completion token count is 11.42M, resulting in a total cost of 11.7 USD.Processing 60K MCTS on GPT-4o-mini costs 47.6 USD.Detailed comparison for synthesis cost are listed below. The GPU price data are from vast​.​ai.
>
> | Method | GPU Type | GPU Hours | Price per hour(USD)  | API Cost（USD） | Total Price(USD) |
> |:---:|:---:|:---:|:---:|:---:|:---:|
> | DART-Math | A100 | 3840 | 0.9 | – | 3456 |
> | SIGMA(GPT4o-mini as Crtique and Revision model) | RTX 4090 | 168 | 0.35 | 47.6 | 106.4 |
> | SIGMA(Qwen2.5-7B-Instructas as Crtique and Revision model) | RTX 4090 | 200 | 0.35 | – | 70 |
>
> Compared to alternative data generation pipelines, SIGMA offers a competitive balance of low computational cost, high data efficiency, and improved overall accuracy. We will include detailed computational cost analysis in the camera-ready version.
> **Final Note：** We sincerely thank the reviewer for the thoughtful and constructive comments. We have revised the paper to clarify the novelty, mechanisms, and cost-efficiency of SIGMA, and hope the additional discussions address your concerns. If our responses have addressed your concerns, please kindly consider raising the score.

---

> > ### Comment · Reviewer_5rCb · 2025-08-08
> >
> > Dear Authors,
> >    Thanks for your rebuttal, after reading your rebuttal, I still feel that "a formal theoretical analysis of why the proposed method works and its potential limitations. A deeper understanding of the underlying mechanisms and potential failure modes would strengthen the paper", this is really necessary and essential for a paper to be accepted.
> >
> > best

---

> > > ### Author Response · Authors · 2025-08-08
> > >
> > > We appreciate the reviewer again the valuable feedback. In response, we provide a formal analysis of SIGMA.
> > >
> > > ### Formal Analysis of the Underlying Mechanism
> > >
> > > Let the optimal reasoning path be denoted by:
> > > $$
> > > T = \\{p^{(1)}, p^{(2)}, \ldots, p^{(D)}\\}
> > > $$
> > > where each $p^{(d)}$ is a textual reasoning step generated by a language model ${M}$. Let $S(p^{(d)})$ denote the set of sibling nodes of step $p^{(d)}$, i.e., alternative generations from ${M}$ conditioned on the same prefix:
> > > $$
> > > S(p^{(d)}) = \\{s_1^{(d)}, s_2^{(d)}, \ldots, s_K^{(d)}\\}, \quad s_k^{(d)} \sim {M}(\cdot \mid p^{(<d)})
> > > $$
> > >
> > > Since all $s_k^{(d)}$ and $p^{(d)}$ are generated from the same model ${M}$ under the same conditions, they are i.d. samples from a shared distribution:
> > > $$
> > > p_\theta(x \mid p^{(<d)})
> > > $$
> > >
> > > where $p^{(<d)}$ denotes the preceding partial reasoning path $p^{(<d)} = \{p^{(1)}, \ldots, p^{(d-1)}\}$. This implies that the **variance** of the distribution across sibling nodes and the selected node is relatively low:
> > > $$
> > > \operatorname{Var}_{x \sim {M}} [f(x \mid p^{(<d)})] \text{ is small}
> > > $$
> > >
> > >
> > > However，each $S(p^{(d)})$ contains latent signals regarding better reasoning trajectories, due to the stochastic nature of sampling and the over-dispersed exploration of ${M}$. To leverage these signals, we introduce a **critique model** ${C}_{\text{LLM}}$, which acts as an information extractor:
> > >
> > >
> > > $$
> > > \Delta(p^{(d)}) := {C}_{\text{LLM}}\big(p_d^*, \{p_d^{(s)} \mid s \in S(p)\}\big) \in \mathbb{R}^k,
> > > $$
> > >
> > >
> > >
> > > Unlike conventional methods comparing each $p^{(d)}$ to a fixed label, here $\Delta(p^{(d)})$ is computed by contrasting $p^{(d)}$ with all siblings in $S(p)$ sharing the same parent. This ensures the relevance of nodes in the loss and yields significantly richer supervisory signals than standard approaches.
> > >
> > > We define a **compressed path representation** $\tilde{T}_p$ as the trajectory obtained after injecting these critique-informed refinements at each step:
> > > $$
> > > \tilde{T}_p = \\{p^{(1)} + \Delta(p^{(1)}), p^{(2)} + \Delta(p^{(2)}), \ldots, p^{(D)} + \Delta(p^{(D)})\\}
> > > $$
> > >
> > > Here, $\tilde{T}_p$ is a functionally transformed version of $T$, where each step integrates feedback from its sibling distribution. Because the critique model aggregates over multiple i.i.d. samples, the refinement $\Delta(p^{(d)})$ serves as a **low-variance estimator** of improvement directions:
> > > $$
> > > \operatorname{Var}[\Delta(p^{(d)})] \ll \operatorname{Var}[s_k^{(d)}] \quad \text{(across } k)
> > > $$
> > >
> > > Therefore, the entire compressed path $\tilde{T}_p$ is formed by:
> > > - sampling from a **shared low-variance source** ($M$)
> > > - aggregating across **multiple sibling nodes**
> > > - filtering through a **deterministic and consistent extractor** $\mathcal{C}_{\text{LLM}}$
> > >
> > > This yields a **dataset of trajectories** $\\{\tilde{T}\_p^i\\}\_{i=1}^N$ with **lower variance**, as each $\tilde{T}\_p^i$ integrates structured, model-consistent feedback from multiple stochastic samples:
> > > $$
> > > \operatorname{Var}_i[\tilde{T}\_p^i] \ll \operatorname{Var}_i[T^i]
> > > $$
> > >
> > >
> > >
> > >
> > > **Theoretical Insight**
> > > To analyze the expected population risk, given a model  $h$ learning from a training set $X$, but evaluated on a test set $X'$ drawn from the data distribution $p_D$. A conceptually useful way to understand this quantity is to decompose it into three components:
> > >
> > > $$
> > > \mathbb{E}_{X^\prime \sim p_D} (R\_{h(X)}(X^{\prime}))= \underbrace{R\_\*(X)}\_{\text {Irreducible Error } E} + \underbrace{R\_{h(X)}(X)-R\_\*(X)}\_{\text{ApproximationGap } A}+ \underbrace{\mathbb{E}\_{X^\prime \sim p_D}(R\_{h(X)}(X^{\prime}))-R\_{h(X)}(X)}\_{\text {Generalization Gap } G} .
> > > $$
> > >
> > > In [1], the generalization  upper bound is related to $\Sigma$, and the $\Sigma$ can be viewed as a variance term that governs the generalization behavior.  A smaller $\Sigma$ leads to a tighter generalization bound. Our method SIGMA could reduces variance via sibling-contrast stabilization, thus tightening the bound, and improving SIGMA series models'generalization.
> > >
> > > ### Failure Modes of Critique-Based Compression
> > >
> > > Although critique-based compression can reduce variance, two risks may counteract this effect:
> > > 1. **Hallucination** — If $\mathcal{C}_{\text{LLM}}$ produces content not grounded in siblings $S(p^{(d)})$, the refinement $\Delta(p^{(d)})$ may deviate from $\mathcal{M}(\cdot \mid p^{(<d)})$, introducing off-distribution noise and increasing variance.
> > > 2. **Low Extraction Capacity** — If $\mathcal{C}_{\text{LLM}}$ fails to distill useful signals, $\Delta(p^{(d)}) \approx 0$ and $F_p \approx T$, yielding minimal compression and no variance reduction.
> > >
> > > Both issues highlight the need for a reliable and well-aligned critique model to ensure effective variance reduction.
> > >
> > >
> > > [1]Finzi, Marc Anton, et al. "Compute-Optimal LLMs Provably Generalize Better with Scale." The Thirteenth International Conference on Learning Representations(ICLR), 2025.

---

> > > ### Author Response · Authors · 2025-08-09
> > > **Follow-up and Appreciation for Your Valuable Feedback**
> > >
> > > Dear Reviewer 5rCb,
> > >
> > > We sincerely thank you for your detailed and thoughtful review of our submission. Your feedback was both precise and insightful, helping us to better understand how our work can be further improved and clarified. We truly appreciate the care and attention you devoted to evaluating our paper.
> > >
> > > Your comments have not only clarified crucial aspects of our research but have also inspired us to refine our explanations and further strengthen the presentation of our contributions. During the rebuttal phase, we have worked diligently to address every point you raised, and your guidance has unquestionably elevated the rigor, clarity, and overall quality of our paper. We view your input as an invaluable contribution to our work’s development.
> > >
> > > If you happen to have any additional questions, thoughts, or suggestions, we would be more than delighted to engage further during the remaining rebuttal period. Your perspective is of great importance to us, and we genuinely welcome any opportunity to benefit from your expertise once again.
> > >
> > > With our sincerest appreciation,
> > >
> > > Authors

---

> ### Author Response · Authors · 2025-08-05
>
> Dear Reviewer 5rCb,
>
> Thank you very much for your detailed and critical feedback. We sincerely appreciate the time and effort you put into reviewing our submission. Your comments have raised important points that prompted us to re-express and improve several aspects of our work.
>
> We have carefully considered your suggestions and made corresponding revisions and clarifications during the rebuttal phase. These changes will also be reflected in the final version of the paper. We believe your input has significantly contributed to enhancing the quality and clarity of our work.
>
> If you have any further questions or concerns, we would be grateful for the opportunity to continue the discussion during the remainder of the rebuttal period.
>
> Best regards,
>
> Authors

---

### Official Review · Reviewer_Aiin · 2025-07-07

**Clarity:** 3
**Significance:** 3
**Originality:** 3
**Rating:** 5
**Confidence:** 3

**Summary:**

This paper introduces SIGMA (SIbling-Guided Monte Carlo Augmentation), a novel framework designed to enhance the reasoning capabilities of large language models (LLMs) by refining Monte Carlo Tree Search (MCTS)-generated reasoning paths. Traditional MCTS-based approaches select a single high-reward reasoning path and discard alternative (sibling) paths, potentially losing valuable information. SIGMA addresses this by:

1. Reusing sibling nodes from the MCTS tree to generate natural language critiques of the selected reasoning steps.

2. Using these critiques as symbolic gradients to iteratively refine the reasoning path via a Textual Gradient Descent (TGD) process.

3. Employing a critique model and a revision model (both LLMs) to perform this refinement without requiring additional rollouts or external supervision.

The authors demonstrate that SIGMA significantly improves the quality of reasoning traces and downstream performance on mathematical reasoning benchmarks, outperforming larger datasets and state-of-the-art baselines with fewer training examples.

**Questions:**

1. What is $n$ & $k$ as mentioned in Section 4.1 in the context of the algorithm given in Section 3? There seems to be some notation abuse here since in Section 3 $n$ is used to denote node and in Section 4.1 it is used to denote candidate completions.

2. In line 280 you say, "Despite sharing the same generator model...". How is the generator model the same for black-box CoT generation and fine-tuning using SIGMA when the former uses GPT-4o-mini (line 279) and the latter uses Qwen2.5-Math-7B?

**Ethical Concerns:**

["NO or VERY MINOR ethics concerns only"]

**Final Justification:**

My main concern with this paper was about the refinement of the reasoning path based on the unselected siblings because it seemed to me that the approach was going to bring the step closer to the unselected siblings. However, the authors have explained that this will not happened as the LLMs used for critiquing and refining the step are prompted to incorporate the benefits of the alternate reasoning paths and to learn from the mistakes in them. On re-reading the relevant section of the paper I am also convinced about this.

The authors have also responded to my other concerns by adding additional results for non-mathematical tasks, pointing me to the results for their approach with large datasets and providing clarifications where the explanation in the paper was not clear.

While I am still not 100% convinced that this method will always be better than simply using more data, I believe it is an important contribution that will be especially useful in settings where data is scarce. Therefore, I am increasing my rating to Accept.

**Limitations:**

Yes

**Quality:**

2

**Strengths And Weaknesses:**

### Strengths

**Quality**: The method is well-motivated and addresses a clear inefficiency in current MCTS-based reasoning pipelines. Empirical results show consistent and significant improvements across multiple benchmarks and model backbones (DeepSeekMath-7B, LLaMA3-8B, Mistral-7B).

**Clarity**: The paper is clearly written, with intuitive figures (e.g., Figures 2–4) that illustrate the SIGMA pipeline. The methodology is well-structured, with detailed explanations of each component (MCTS, critique generation, revision).

**Significance**: SIGMA offers a data-efficient alternative to large-scale dataset construction, which is valuable given the high cost of data generation. The framework is model-agnostic and could be extended to other domains beyond math reasoning.

**Originality**: The idea of treating sibling nodes as sources of symbolic feedback and using them for gradient-like refinement is novel to the best of my knowledge.

### Weaknesses:

1. The key point about refining the selected step $T_p$ based on the unselected siblings $T_s$ feels a little counterintuitive. If the siblings were not selected because they were expected to yield inferior results by the MCTS process then why would refining the content of $T_p$ to be closer to the content of $T_s$ be beneficial?

2. The experiments are restricted to mathematical reasoning tasks. While the authors claim generalizability, this is not empirically validated.

3. Given the extra effort involved in preparing the data using SIGMA and the fact that MathFusion-60K gives better results for two (LLaMA3 & Mistral) out of the three models considered, it isn't clear if we should go with SIGMA or just use a larger dataset.

---

> ### Author Rebuttal · Authors · 2025-07-31
>
> We appreciate the reviewer for the thoughtful feedback，recognizing (i) data-efficiency for dataset construction (ii) well-structured methodology.We've addressed your concerns below.
> > The key point about refining the selected step $T_p$ based on the unselected siblings $T_s$ feels a little counterintuitive. If the siblings were not selected because they were expected to yield inferior results by the MCTS process then why would refining the content of $T_p$ to be closer to the content of $T_s$ be beneficial?
>
> The central contribution of our work lies in how to extract constructive revision signals for the selected step $T_p$ from its suboptimal siblings $T_s$. Our method does not refine the content of $T_p$ to be closer to the content of $T_s$. Instead, we first apply a critique model to analyze $T_s$, and then use a revision model to improve $T_p$ based on the critique outcomes.
>
> This design enables two key benefits:
> – Alternative reasoning paths in $T_s$ are leveraged to enrich the revision of $T_p$ without copying them directly.
> – Mistakes in $T_s$ are explicitly identified and used to detect and correct similar issues in $T_p$, rather than introducing the same errors into $T_p$.
>
> We will clarify these mechanisms in the camera-ready version.
> > The experiments are restricted to mathematical reasoning tasks. While the authors claim generalizability, this is not empirically validated.
>
> We appreciate the reviewer for the helpful suggestion.We agree that evaluating the generalizability of SIGMA beyond mathematical reasoning is important. Owing to its design—which relies entirely on textual signals for refining reasoning paths—SIGMA holds strong potential in non-mathematical domains as well.
>
> Following the experimental setup of RevThink[1], we applied SIGMA to CommonsenseQA, StrategyQA, and ARC-Challenge. We conducted LoRA finetuning on Mistral-7B-Instruct-v0.3 and Gemma-7B-Instruct to assess SIGMA’s cross-domain generalization ability. SIGMA outperformed RevThink and other baselines, achieving state-of-the-art performance(75.74 AVG on Mistral-7B-Instruct-v0.3 and 73.63 AVG on Gemma-7B-Instruct) on these benchmarks.
>
> Detailed results are provided in the table below.
>
> | Methods | Mistral-7B-Instruct-v0.3 |  |  |  | Gemma-7B-Instruct |  |  |  |
> |:---:|:---:|:---:|:---:|:---:|:---:|:---:|:---:|:---:|
> |  | SQA | CSQA | ARC | AVG | SQA | CSQA | ARC | AVG |
> | Zero-shot | 53.89 | 62.57 | 73.68 | 63.38 | 56.33 | 66.26 | 68.34 | 63.64 |
> | SKD | 63.76 | 71.86 | 74.66 | 70.09 | 56.77 | 72.48 | 73.29 | 67.51 |
> | Distill Step-by-Step | 64.19 | 71.92 | 75.32 | 70.48 | 56.77 | 73.01 | 72.92 | 67.57 |
> | Rephrase Question | 65.07 | 70.19 | 74.51 | 69.92 | 54.15 | 70.22 | 72.37 | 65.58 |
> | Question Aug | 65.07 | 72.23 | 73.32 | 70.21 | 57.21 | 68.11 | 72.74 | 66.02 |
> | Answer Aug | 66.38 | 69.12 | 76.77 | 70.76 | 57.21 | 73.01 | 73.92 | 68.05 |
> | Revthink | 70.97 | 75.76 | 78.5 | 75.08 | 64.19 | 74.53 | 75.09 | 71.27 |
> | SIGMA(Ours) | 71.62 | 76.3 | 79.3 | 75.74 | 67.4 | 75.4 | 78.1 | 73.63 |
>
> We will include the discussion on generalization and the corresponding experimental results in the camera-ready version.
> > Given the extra effort involved in preparing the data using SIGMA and the fact that MathFusion-60K gives better results for two (LLaMA3 & Mistral) out of the three models considered, it isn't clear if we should go with SIGMA or just use a larger dataset.
>
> The scalability of SIGMA to larger datasets is essential to its general applicability. In the main text, we emphasized its data efficiency by demonstrating that SIGMA achieves comparable or even superior performance with only half or fewer training samples compared to other methods, and in Table 1 we present results comparing the SIGMA-30K series against the 60K MathFusion dataset to further illustrate this point.
>
> To further address this concern, we include in Appendix A.2 additional results of SIGMA trained on the full 60K dataset. SIGMA continues to outperform all baselines under this setting, achieving state-of-the-art results. This provides important empirical evidence of its effectiveness at scale.
>
> We will incorporate this finding into the camera-ready version. We appreciate the reviewer for the valuable suggestion.
>
> > What is $n$ & $k$ as mentioned in Section 4.1 in the context of the algorithm given in Section 3? There seems to be some notation abuse here since in Section 3 $n$ is used to denote node and in Section 4.1 it is used to denote candidate completions.
>
> In Section 4.1, the setting of $k = 5$ and $n = 3$ refers to the sampling of 5 candidate outputs for each non-terminal MCTS node, from which the top 3 are selected as its child nodes. This is consistent with the use of $n$ in Section 3, where it denotes a node in the MCTS tree. The term "candidates" in Section 4.1 refers to possible nodes considered for inclusion in the best trajectory.
>
> We appreciate the reviewer for pointing out this ambiguity. We will revise the notation and clarify the explanation in the camera-ready version to avoid confusion.
> >In line 280 you say, "Despite sharing the same generator model...". How is the generator model the same for black-box CoT generation and fine-tuning using SIGMA when the former uses GPT-4o-mini (line 279) and the latter uses Qwen2.5-Math-7B?
>
> In Section 4.4, we conducted ablation studies that mesures the contribution of refinement by testing models fine-tuned by datasets with same problems sets but different answer synthesis methods.The results are shown in Table 2.
>  - Lines with suffix "SIGMA":Qwen2.5-Math-7B is used to generate the original MCTS.GPT-4o-mini is used to conduct the refinement.
>  - Lines with suffix "MCTS":Qwen2.5-Math-7B is used to generate the original MCTS.For each question,extract best tracjectroy based on Q-value and discard all other nodes.
>  - Lines with suffix "Blackbox":For each question,generate step-by-step answer with standard COT prompt by GPT-4o-mini-2024-07-18.
>
> The results proves that (i)Improvement of refinement(SIGMA vs MCTS) (ii)Refine from siblings outperforms oracle model(SIGMA vs Blackbox)
>
> The statement “Despite sharing the same generator model...” was intended to highlight the contrast between SIGMA and the Blackbox variant, where SIGMA achieves superior performance despite refining from sibling traces instead of using an oracle CoT generated directly by the same backbone used for refinement.
>
> We appreciate the reviewer’s correction and will revise the wording in the camera-ready version to improve clarity.
>
> [1]Chen, J. C. Y., Wang, Z., Palangi, H., Han, R., Ebrahimi, S., Le, L., ... & Pfister, T. (2024). Reverse thinking makes llms stronger reasoners. arXiv preprint arXiv:2411.19865.
>
> **Final Note:** Thank you for your detailed feedback on sibling-based refinement, generalization beyond math, and scalability. We hope our empirical results and clarifications have addressed your concerns and kindly invite you to consider raising your score.

---

> ### Author Response · Authors · 2025-08-05
>
> Dear Reviewer Aiin,
>
> Thank you again for your time and thoughtful review. Your feedback has been very helpful in improving the quality and clarity of our paper, and we are grateful for the points you raised during the review phase.
>
> In the revised version, we will carefully incorporate your suggestions along with the relevant clarifications and results discussed during the rebuttal period. These updates have helped us present our ideas more clearly and rigorously.
>
> Please feel free to reach out if you have any further questions or thoughts. We remain fully engaged and open to additional discussion throughout the remainder of the rebuttal phase.
>
> Best regards,
>
> Authors

---

> > ### Comment · Reviewer_Aiin · 2025-08-06
> > **Score update**
> >
> > Thank you for responding to all my concerns in detail. I appreciate the effort and have updated my score. I do not have any other questions or concerns at this time.

---

> > > ### Author Response · Authors · 2025-08-06
> > >
> > > Dear Reviewer Aiin,
> > >
> > > Thank you very much for your thoughtful follow-up and for taking the time to carefully review both our paper and our rebuttal. We deeply value your insights and the constructive comments you provided during the review process, which were instrumental in helping us improve the quality and clarity of our work.
> > >
> > > We will carefully incorporate the corresponding revisions into the final version of the paper. Your engagement and attention throughout the review are sincerely appreciated.
> > >
> > > Best regards,
> > >
> > > Authors

---

### Official Review · Reviewer_F8sN · 2025-07-08

**Clarity:** 3
**Significance:** 3
**Originality:** 3
**Rating:** 4
**Confidence:** 4

**Summary:**

This paper presents SIGMA, an enhanced MCTS framework that refines model-generated reasoning traces by leveraging sibling information in the search tree. Given a standard MCTS that explores the space of multi-step CoT paths and selects an optimal path, the SIGMA framework additionally refines the CoT on every node of the selected path, by considering the discrepancies between the selected node and discarded sibling node. For every refinement, a critique language model is used to suggest how to revise and a revise language model takes that suggestion and generates an improved version.

The author applied this framework to the math domain, with a setup that prompts and final answers are given, and the goal is to refine the reasoning traces to construct better training data. The training set is based on prompts from MATH and GSM8K datasets, and Qwen2.5-Math-7B is used to generate the original reasoning paths. Then, GPT-4o-mini is used to conduct the refinement. The resulting datasets are compared to existing math datasets of similar size or larger size, demonstrating better performance in terms of training math models.

**Questions:**

- What is the reasoning behind choosing Qwen2.5-math for generating the original CoT and GPT-4o-mini for the refinement?
- For the training experiments, why not finetune Qwen2.5 model using the resulting data? It might be intriguing to see the model's self-improvement using this framework.

**Ethical Concerns:**

["NO or VERY MINOR ethics concerns only"]

**Final Justification:**

I have read the rebuttal and other reviews.

The new results provided in the rebuttal answered my original questions and concerns, making it a sound extension to the MCTS framework, especially for generating targeted CoT data.

With its current technical novelty and contribution, I would like to keep my original recommendation.

**Limitations:**

yes

**Quality:**

3

**Strengths And Weaknesses:**

Strengths:
- The paper addresses a timely topic of enhancing the MCTS framework for generating more effective CoT reasoning paths. The proposed improvement incorporates more information from sibling nodes looks meaningful and promising.
- I found the formulation of the critique model and refine model under the textual gradient descent very intriguing and inspiring. This may be another good attempt of the [textgrad](https://arxiv.org/abs/2406.07496) framework.
- The experiment results demonstrated better performance of the resulting math datasets.

Weaknesses:
- My major complaint is about the evaluation setup. The resulting datasets in this paper is based on CoT reasoning generated with Qwen2.5-Math model and then refined by GPT-4o-mini. The other existing datasets for comparison were generated with different models (e.g., MathFusion was generated using GPT-4o-mini only?). I believe that the base model makes a considerable difference here, and it's not clear to me how the proposed refinement contributed to the final improvement.

---

> ### Author Rebuttal · Authors · 2025-07-31
>
> We appreciate the reviewer for the thoughtful feedback，recognizing (i) novelty of incorporates more information from sibling nodes (ii) inspring attempt that formulating critique model and refine model based on textual gradient descent. We've addressed your concerns below.
> > The other existing datasets for comparison were generated with different models (e.g., MathFusion was generated using GPT-4o-mini only?).I believe that the base model makes a considerable difference here, and it's not clear to me how the proposed refinement contributed to the final improvement.
>
> Thank you for your attention to the impact of the generation model and the effectiveness of the proposed refinement. I address this concern below.
>
> Switching refinement model to weaker open-source LLMs resultd in a comparable performance compared with closed-source model as refinment models.Under Qwen-2.5-72B-instruct，the results achieves 34.3 on Llama3-8B，which is still comparable；31 on mistral，and 46.5 on deepseekMath-7B,which remains state-of-the-art with other methods.
>
> | **15K Samples** | MATH | GSM8K | College | DM | Olympiad | Theorem | AVG |
> |:---:|:---:|:---:|:---:|:---:|:---:|:---:|:---:|
> | **Llama3-8B Series** |  |  |  |  |  |  |  |
> | Qwen2.5-7B-Instruct as Critique and Revision Model | 35.3  | 77.1  | 21.7  | 39.5  | 9.6  | 16.9  | 33.3  |
> | Qwen2.5-72B-Instruct as Critique and Revision Model | 36.3  | 78.9  | 23.2  | 39.4  | 9.8  | 18.0  | 34.3  |
> | GPT4o-mini as Critique and Revision Model | 36 | 82 | 24.2 | 42 | 10.5 | 22 | 36.1 |
> | MathFusion（30K Average Score） | 37.2  | 76.7  | 23.9  | 37.1  | 12.1  | 17.1  | 34.1  |
> | **Mistral-7B-v0.1 Series** |  |  |  |  |  |  |  |
> | Qwen2.5-7B-Instruct as Critique and Revision Model | 28.8 | 79.3 | 18.4 | 33.3 | 6.7 | 13.5 | 30 |
> | Qwen2.5-72B-Instruct as Critique and Revision Model | 30.6 | 76.8 | 21.1 | 35.3 | 6.8 | 15.6 | 31 |
> | GPT4o-mini as Critique and Revision Model | 30 | 75.3 | 20.8 | 39.5 | 7.7 | 16.3 | 31.6 |
> | MathFusion（30K Average Score） | 30.0  | 74.0  | 18.5  | 25.7  | 9.2  | 14.5  | 28.6  |
> | **DeepSeekMath-7B-Base Series** |  |  |  |  |  |  |  |
> | Qwen2.5-7B-Instruct as Critique and Revision Model | 49.8  | 80.6  | 38.3  | 60.6  | 18.4  | 24.1  | 45.3  |
> | Qwen2.5-72B-Instruct as Critique and Revision Model | 51.3  | 81.9  | 39.9  | 64.8  | 17.5  | 23.8  | 46.5  |
> | GPT4o-mini as Critique and Revision Model | 52.2 | 81.1 | 37.4 | 64.5 | 20.3 | 26.1 | 47 |
> | MathFusion（30K Average Score） | 49.8  | 76.0  | 38.2  | 60.7  | 20.0  | 21.9  | 44.4  |
> | **Qwen2.5-Math-7B Series** |  |  |  |  |  |  |  |
> | Qwen2.5-7B-Instruct as Critique and Revision Model | 74.0  | 89.9  | 48.5  | 83.1  | 39.3  | 40.6  | 62.6  |
> | Qwen2.5-72B-Instruct as Critique and Revision Model | 76.9  | 91.2  | 48.6  | 87.0  | 39.6  | 39.0  | 63.7  |
> | GPT4o-mini as Critique and Revision Model | 79.9  | 89.2  | 45.2  | 88.8  | 43.4  | 47.0  | 65.6  |
>
> In Section 4.4, we conducted ablation studies that mesures the contribution of refinement by testing models fine-tuned by datasets with same problems sets but different answer synthesis methods.The results are shown in Table 2 at page 9 of our paper.
>  - Lines with suffix "SIGMA":Qwen2.5-Math-7B is used to generate the original MCTS.GPT-4o-mini is used to conduct the refinement.
>  - Lines with suffix "MCTS":Qwen2.5-Math-7B is used to generate the original MCTS.For each question,extract best tracjectroy based on Q-value and discard all other nodes.
>  - Lines with suffix "Blackbox":For each question,generate step-by-step answer with standard COT prompt by GPT-4o-mini-2024-07-18.
>
> The results proves that (i)Improvement of refinement(SIGMA vs MCTS) (ii)Refine from siblings outperforms oracle model(SIGMA vs Blackbox)
>
> > What is the reasoning behind choosing Qwen2.5-math for generating the original CoT and GPT-4o-mini for the refinement?
>
> Thank you for the insightful question. Our method is not restricted to specific choices of generation or refinement models. A core focus of our work is data efficiency. In early-stage testing, we experimented with several math-specialized models including Mathstral and DeepSeek-Math. We ultimately selected Qwen2.5-Math-7B for CoT generation due to its strong MCTS generation speed and token efficiency,that it produces fewer tokens per problem on average,reducing the overall computational cost of both generation and refinement.
>
> For the refinement model, GPT-4o-mini was chosen as a cost-effective option. However, our method does not depend on closed-source models. We have shown performance on different critique models above.
>
> We will clarify these design choices and their implications in the camera-ready version.
> > For the training experiments, why not finetune Qwen2.5 model using the resulting data? It might be intriguing to see the model's self-improvement using this framework.
>
> Thank you for your valuable suggestion. In the main experiments, we did not finetune Qwen2.5-Math-7B because existing strong baselines such as DART and MathFusion and other baselines have not reported results on this model. To ensure fair and verifiable comparisons, we focused on base models with publicly available results.
>
> To further assess the self-improvement ability of our method, we additionally conducted finetuning experiments on Qwen2.5-Math-7B.SIGMA remains state-of-the-art performace on Qwen2.5-Math-7B, reaching 79.92 for MATH, 89.23 for GSM8K and 65.59 on all datasets average,surpassing other methods such as Eurus and OpenMath.The results are shown in the following table.
> | Qwen2.5-Math-7B (Fine-tuned) | # Samples | MATH | GSM8K | College | DM | Olympiad | Theorem | AVG |
> |:---:|:---:|:---:|:---:|:---:|:---:|:---:|:---:|:---:|
> | Rstar@Greedy | 747K | 78.40  | 89.70  | **52.50** | – | **47.10** | – | – |
> | Eurus-2-7B-PRIME | 230K | 53.88  | 42.99  | 20.97  | 47.30  | 22.22  | 18.62  | 34.33  |
> | OpenMath-Nemotron-7B | 3.2M | 40.52  | 73.39  | 32.93  | 54.70  | 11.41  | 13.63  | 37.76  |
> | qwen2.5math-instruct | 2.5M | 82.16  | 95.15  | 46.80  | 82.00  | 41.60  | 40.75  | 64.74  |
> | AceMath-7B-Instruct | 2.3M | 75.40  | 58.38  | 45.82  | 59.70  | 39.85  | 30.75  | 51.65  |
> | DART-Math-60K | 60K | 63.14  | 91.28  | 47.23  | 74.90  | 26.81  | 35.38  | 56.46  |
> | MathFusion-60K | 60K | 56.14  | 59.59  | 37.08  | 67.70  | 41.19  | 27.87  | 48.26  |
> | SIGMA-60K | 60K | 79.92  | 89.23  | 45.17  | **88.80** | 43.41  | **47.00** | **65.59** |
>
> **Final Note:** Thank you for your thoughtful feedback on refinement effectiveness, model variation, and self-improvement on Qwen2.5. We hope our clarifications and results have addressed your concerns, and would appreciate your consideration in raising the score.

---

> > ### Comment · Reviewer_F8sN · 2025-08-04
> >
> > I appreciate the detailed rebuttal from the authors and the additional experiments. The results answered my questions and concerns, making it a sound extension to the MCTS framework, especially for generating targeted CoT data. With its current technical novelty and contribution, I would like to keep my score.

---

> > > ### Author Response · Authors · 2025-08-05
> > >
> > > Dear Reviewer F8sN,
> > >
> > > Thank you for your careful reading and for your feedback on our rebuttal. Your comments were very helpful in improving our work. We will incorporate the relevant revisions and updates into the final version of the paper. We sincerely appreciate your time and efforts throughout the review process.
> > >
> > >
> > > Best regards,
> > >
> > > Authors

---

### Note · Authors · 2025-08-13

First, we sincerely thank the AC and all five reviewers (**F8sN**, **Aiin**, **Jhvz**, **UihT**, **5rCb**) for their thoughtful feedback. Below is a concise summary of what we added or clarified during rebuttal, following their suggestions.

---

## Additional Experiments

- **Reviewer F8sN and Jhvz**: We fine-tuned **Qwen2.5-Math-7B** with **SIGMA**; it achieves **65.59 avg** on the 60K setting, outperforming prior methods including **AceMath and Rstar**.
- **Reviewers F8sN, Jhvz & UihT**: We replaced the critique/revision model with open-source alternatives (**Qwen2.5-7B/72B**) and found comparable trends to **GPT-4o-mini**.Our experiments also found that self-distillation using a 7B model alone can achieve strong performance.
- **Reviewer Jhvz**: We studied **sibling-count sensitivity** and observed **monotonic gains** as the number of siblings increases.
-  **Reviewers Aiin, Jhvz& UihT**: Following RevThink’s experimental setup, we applied SIGMA to **CSQA**, **StrategyQA**, and **ARC-Challenge**, using LoRA fine-tuning on **Mistral-7B-Instruct** and **Gemma-7B-Instruct**. SIGMA consistently outperformed RevThink and other baselines, demonstrating the **generalization ability of our framework across domains**.
- **Reviewer UihT and 5rCb**: We added **compute/token-efficiency analyses** and a **cost table**. We showed that SIGMA is cost-efficient, and further demonstrated that replacing GPT-4o-mini with open-source models can significantly reduce costs with only a minimal drop in performance.
---

## Clarifications and Analysis

- **Reviewer 5rCb**: We added a **formal analysis** explaining why **sibling-guided refinement reduces variance** and discussed **failure modes**.
- **Reviewers UihT & Jhvz & Aiin**: We clarified the **mechanism**: **SIGMA does not copy sibling content**; it **contrasts selected steps with siblings** to extract targeted critique before revision.

---
**Overall**, **SIGMA** showcases strong and consistent generalization ability while delivering clear cost advantages, including outstanding token efficiency and overall computational economy.

We are sincerely grateful to the **AC** for the thoughtful coordination and to all **reviewers** for their constructive, detailed feedback and valuable time. Your insights have helped us refine both our presentation and our technical understanding. We will incorporate these clarifications and improvements into the revised manuscript to ensure it fully addresses your comments and suggestions.

---

### Decision · Program_Chairs · 2025-09-17

**Decision:**

Accept (poster)

**Comment:**

This paper introduces SIGMA, a data-efficient framework for refining chain-of-thought (CoT) reasoning paths generated by Monte Carlo Tree Search (MCTS). The main contribution is a two-stage refinement process: instead of discarding non-optimal sibling nodes in the search tree, leveraging them to provide corrective feedback. A "critique" model first identifies the strength and weakness of the chosen reasoning step relative to its siblings, and generate a "symbolic gradient" in natural language. A "revision" model then applies this feedback to improve the original step. This method systematically extracts valuable signals from the entire search tree, significantly enhancing the quality of the final training data. The work is relevant to NeurIPS, addressing the critical challenge of improving data quality for complex LLM reasoning tasks.

Most reviewers were positive about the paper, with scores leaning towards acceptance: 5, 5, 5, 4, 3.
Also, confidence level is generally high due to the clarity of the proposed method and the strength of the empirical results.

Reviewers pointed out the novelty and intuition of the core idea (reusing discarded sibling nodes). The data efficiency demonstrated in the experiments is quite impressive and highlighted as a major strength. The comprehensive evaluation across multiple base models and benchmarks was also commended.

One concern raised is this framework relies on a capable proprietary model (GPT-4o-mini) for the critique and revision steps. This dependency could limit the accessibility of the method and raises questions about whether the performance gains are partly attributable to the knowledge transfer from this "teacher" model.

The method is presented as a heuristic approach. The analogy to "Textual Gradient Descent" is intuitive but not theoretically rigorous. However, the overall methodology is sound, well-motivated, and clearly described.

The empirical evidence is a key strength of this work - the authors show that models trained on small, SIGMA-refined datasets (15K-30K samples) consistently outperform models trained on much larger datasets from prior work. The well-designed ablation studies effectively isolate the benefits of the sibling-guided refinement process over both vanilla MCTS paths and standard black-box CoT generation.

Furthermore, the paper provides sufficient detail regarding the experimental setup, models, and hyperparameters, suggesting that the results are reproducible. This work has the potential for significant impact on how the community approaches data synthesis for LLMs. By demonstrating that substantial gains can be achieved by more intelligently processing the outputs of existing generation methods, this paper offers a compelling alternative to simply scaling dataset size.

Overall, this is a nice paper with a novel, well-executed idea and compelling empirical results. The method's demonstrated ability to create high-quality training data with remarkable efficiency is a significant contribution. While the reliance on a proprietary teacher model is a valid concern, the novelty of the framework and the strength of the results make it a clear accept. The work will be of great interest to the NeurIPS community and is well-suited for discussion in a poster session.